# *GaussianLens*: Localized High-Resolution Reconstruction via On-Demand Gaussian Densification

## ABSTRACT

We perceive our surrounding environments with an active focus, paying more attention to regions of interest, such as the shelf labels in a grocery store or a family photo on the wall. When it comes to scene reconstruction, this human perception trait calls for spatially varying degrees of detail ready for closer inspection in critical regions, preferably reconstructed on demand as users shift their focus. While recent approaches in 3D Gaussian Splatting (3DGS) can achieve fast, generalizable scene reconstruction from sparse views, their uniform resolution output leads to high computational costs, making them unscalable to high-resolution training. As a result, they cannot leverage available image captures at their original high resolution for detail reconstruction. Per-scene optimization methods reconstruct finer details with heuristic-based adaptive density control, yet require dense observations and lengthy offline optimization. To bridge the gap between the prohibitive cost of high-resolution holistic reconstructions and the user needs for localized fine details, we propose the problem of localized high-resolution reconstruction through on-demand generalizable Gaussian densification. Given an initial low-resolution 3DGS reconstruction, the goal is to learn a generalizable network that densifies the reconstruction to capture fine details in a user-specified local region of interest (RoI), based on sparse high-resolution observations of the RoI. This formulation avoids the high cost and redundancy of uniformly high-resolution reconstructions and enables the full leverage of high-resolution observations in critical regions. To address the problem, we propose *GaussianLens*, a feed-forward densification framework that fuses multi-modal information from the initial 3DGS and multi-view images. We further propose a pixel-guided densification mechanism that effectively captures details under significant resolution increases. Experiments demonstrate our method's superior performance in local high-fidelity detail reconstruction and strong scalability to images of up to $1024 \times 1024$ resolution.

## 1 INTRODUCTION

3D Gaussian Splatting (3DGS) (Kerbl et al., 2023; Huang et al., 2024; Yu et al., 2024c) has shown great promise in photorealistic 3D reconstruction and novel-view synthesis. More recently, generalizable 3DGS reconstruction methods (Charatan et al., 2024; Wewer et al., 2024; Szymanowicz et al., 2024b;a; Tang et al., 2025; Xu et al., 2024b; Zhang et al., 2025b;a; Chen et al., 2025a) have extended these capabilities to sparse input views and on-the-fly reconstruction settings. Most existing works predict Gaussians with a regular structure, typically pixel-aligned (Charatan et al., 2024; Szymanowicz et al., 2024b; Chen et al., 2025b; Xu et al., 2024a; Tang et al., 2025; Zhang et al., 2025b) or voxel-aligned (Chen et al., 2025a; Zhang et al., 2024b). While structured outputs facilitate learning, they lead to a uniform-resolution reconstruction, with the same number of Gaussians allocated to each pixel or voxel. This approach is inefficient, as capturing fine details requires a computationally costly global increase in the reconstruction resolution, making it impractical to leverage readily available high-resolution observations. For example, while DL3DV (Ling et al., 2024) contains $3840 \times 2160$ videos, existing novel-view synthesis works only use up to $960 \times 540$, with many defaulting to $256 \times 256$. Training state-of-the-art DepthSplat (Xu et al., 2024a) on $1024 \times 1024$ inputs is also infeasible as it cannot fit into the memory of an 80GB H100 GPU. In contrast, per-scene

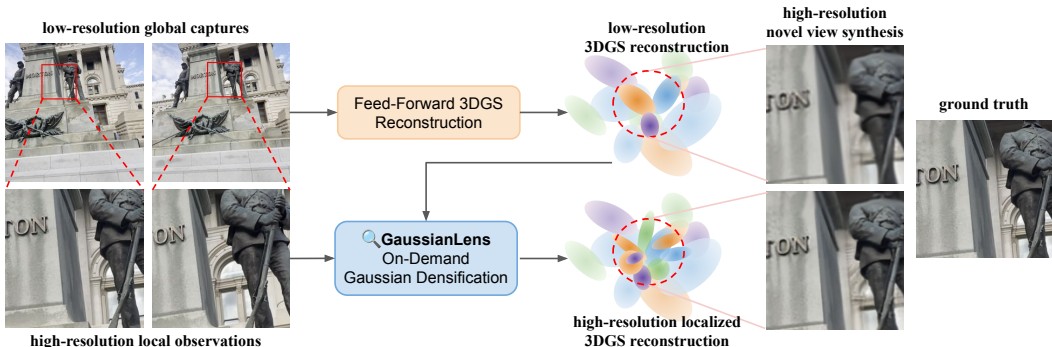

Figure 1: We introduce the problem of localized high-resolution reconstruction via on-demand Gaussian densification. While the majority of feed-forward models are confined to single-pass, uniform-resolution reconstruction, *GaussianLens* achieves low-cost, high-resolution local reconstruction by learning to densify low-resolution initial 3DGS reconstructions conditioned on high-resolution local observations.

optimization methods produce non-uniform reconstructions adapted to varying scene complexity via Adaptive Density Control (Kerbl et al., 2023), a heuristic mechanism to densify Gaussians in under-reconstructed regions. Yet they rely on dense observations and lengthy offline optimization. To sum up, existing methods are bottlenecked by computational costs to fully leverage all available high-resolution images for reconstruction, especially in fast, interactive settings.

Meanwhile, uniformly high-resolution reconstruction of the entire scene is often unnecessary. Humans typically focus on the fine details within a small region of interest, while a lower-resolution view of the surroundings suffices for a holistic understanding. For example, in an interactive room capture, the user may want to ensure the titles on book spines are legible, but care less about the tiny dents on the floor. This perceptual trait calls for a more efficient approach: reconstructions with spatially varying degrees of detail, where critical areas are reconstructed at higher fidelity. To reduce redundancy and cost, these details are ideally reconstructed on demand as the user's focus shifts.

To bridge the gap between prohibitively costly high-resolution holistic reconstructions and the user needs for localized fine details, we introduce the problem of localized high-resolution reconstruction through on-demand Gaussian densification. Given a low-resolution 3DGS reconstruction of the entire scene, the goal is to densify a user-specified region of interest to reveal finer details, guided by a sparse set of high-resolution images of the region. The region is specified by 2D masks, mimicking the scenario where a user selects an area from the current view to "zoom in" for more details. The reconstruction is evaluated on high-resolution novel view renderings of the selected region.

To address the problem, we introduce *GaussianLens*, a cross-modal framework that aggregates information from 2D images to the initial 3D Gaussians via complementary mechanisms. We first render the initial 3D Gaussians and compare them to groundtruth observations to obtain residual information, based on which we construct initial Gaussian features and image features. We then use a PointTransformerV3 (Wu et al., 2024)-based encoder for Gaussian feature extraction, where we introduce projection-based image-Gaussian cross-attention layers to further fuse information from images. Finally, we decode Gaussian features into densified Gaussian parameters, expressed as offsets from initial Gaussian parameters. The whole process can be viewed as a learned version of the densify-by-clone step and subsequent optimizations in per-scene 3DGS reconstruction.

However, with larger resolution increases, cloning-based densification struggles to capture all newly introduced details. We therefore propose *pixel-guided densification*, where we spawn one Gaussian for each high-resolution pixel in the region of interest. These Gaussians effectively preserve high-resolution details and complement the input Gaussians that serve as a coarse scaffold.

To summarize, our main contributions are:

- We propose the problem of localized high-resolution reconstruction via on-demand Gaussian densification, a formulation that avoids the prohibitive and unnecessary cost of uniformly high-resolution reconstructions and enables full leverage of high-resolution local observations.
- We develop *GaussianLens*, a multi-modal framework featuring complementary mechanisms to fuse information from 3D Gaussians and multi-view images for effective densification prediction.

- We propose a novel pixel-based densification mechanism that better captures details under substantial resolution increase in the one-step feed-forward densification scenario.

We build a benchmark for the proposed problem based on RealEstate10K (Zhou et al., 2018) and DL3DV (Ling et al., 2024), and compare our method against state-of-the-art generalizable 3DGS reconstruction methods. We achieve efficient on-demand high-resolution detail reconstruction in local regions with qualities above or on par with uniformly high-resolution models, while using fewer computational resources. We can also leverage up to $1024 \times 1024$ high-resolution observations, which uniform feed-forward models cannot scale up to.

## 2 RELATED WORK

**Generalizable 3D Gaussian Prediction** Generalizable 3D Gaussian prediction methods learn to predict 3D Gaussian reconstructions with a network forward pass, typically conditioned on sparse observations. Significant progress has been made to reconstruct objects (Zhang et al., 2025b; Xu et al., 2024b; Zhang et al., 2024b; Chen et al., 2025a; Tang et al., 2025; Szymanowicz et al., 2024b; Zou et al., 2024; Lu et al., 2024; Shen et al., 2024a) and scenes (Charatan et al., 2024; Wewer et al., 2024; Szymanowicz et al., 2024a; Chen et al., 2025b; Xu et al., 2024a; Zhang et al., 2025a; Liu et al., 2025) under sparse-view settings. Integration with large 2D (Blattmann et al., 2023) or geometric (Wang et al., 2024; Leroy et al., 2024) foundation models enables 360° view synthesis (Chen et al., 2024a), semantic (Fan et al., 2024b) and pose-free reconstructions (Kang et al., 2024; Fan et al., 2024a; Li et al., 2025; Ye et al., 2024a; Hong et al., 2024). Most methods predict a structured set of Gaussians, typically pixel-aligned (Charatan et al., 2024; Szymanowicz et al., 2024b; Chen et al., 2025b; Xu et al., 2024a; Tang et al., 2025; Zhang et al., 2025b) or voxel-aligned (Chen et al., 2025a; Zhang et al., 2024b). While structured outputs facilitate learning, their uniform resolution limits their ability to reconstruct high-resolution details due to the high computational cost. An exception, PanSplat (Zhang et al., 2024a), specifically targets 4K ($2048 \times 4096$) panorama synthesis. It supports up to $768 \times 1536$ with efficient hierarchical cost volume and Gaussian head designs, but scaling to 4K is only achieved with deferred backpropagation.

**Generalizable 3D Gaussian Update** More recently, generalizable frameworks have been used to update given initial Gaussians. Chen et al. (2024b) learns to iteratively update Gaussians by leveraging cues from rendering gradients. Chen et al. (2024c) trains a point transformer to refine flawed 3D Gaussians to reduce artifacts at out-of-distribution views. Most related to us is Generative Densification (GD) (Nam et al., 2024), which proposes to attach a learned Gaussian densification module to feed-forward frameworks to improve the reconstruction in high-frequency, under-reconstructed regions. However, GD consumes latent features from the base feed-forward model, and has to be tailored to and fine-tuned with it. The resulting model still performs single-pass image-to-Gaussian prediction for the full scene. In contrast, our model is source-agnostic, operating directly on the Gaussians with no access to or assumptions about the source model. We also have the flexibility to build on an existing reconstruction and densify exclusively in specified local regions.

**Adaptive Density Control** In per-scene 3D Gaussian optimization, Adaptive Density Control (Kerbl et al., 2023) enables efficient reconstruction of spatially varying scene details, by heuristically selecting and densifying Gaussians in under-reconstructed regions. Many methods have been proposed to improve the selection heuristic (Ye et al., 2024b; Zhang et al., 2024c; Rota Bulò et al., 2024; Kheradmand et al., 2024; Lyu et al., 2024; Cheng et al., 2024) and the initialization of new Gaussians (Kheradmand et al., 2024; Rota Bulò et al., 2024; Lyu et al., 2024; Cheng et al., 2024). However, they all require dense observations and time-consuming per-scene optimization.

**Multi-Scale Gaussian Splatting** Multi-scale and hierarchical Gaussian Splatting methods reconstruct the scene with layers of Gaussians capturing scene details at different scales. During rendering, levels of details are flexibly chosen based on computational resources and user needs, allowing large-scale scene reconstruction (Kerbl et al., 2024; Liu et al., 2024a; Ren et al., 2024), anti-aliased view-adaptive rendering (Yan et al., 2024; Shi et al., 2024; Seo et al., 2024; Di Sario et al., 2025), and generalizable coarse-to-fine scene reconstruction (Tang et al., 2024). They require lengthy offline optimization and costly storage of the full Gaussian hierarchy, before *rendering* with flexible level-of-detail. In contrast, we aim to support on-demand level-of-detail at the *reconstruction* stage.

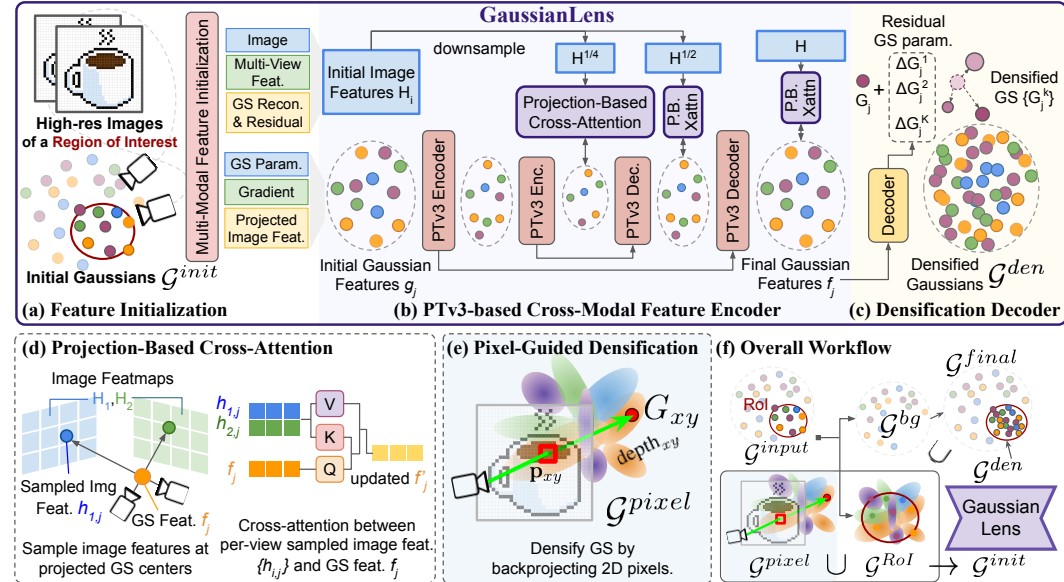

Figure 2: **Method overview**. (a)-(d) illustrates *GaussianLens*, our feed-forward densification framework. It constructs multi-modal features for initial Gaussians and images (a), further extracts features via a PTv3-based encoder with projection-based cross attention (d) to images (b), and decodes them into residual parameters of densified Gaussians (c). (e) illustrates our pixel-guided densification. (f) shows the overall workflow.

**Super-Resolution and Close-Up Novel View Synthesis** Most super-resolution and close-up novel view synthesis methods reconstruct high-resolution details by distilling 2D image super-resolution models (Yoon & Yoon, 2023; Lee et al., 2024; Yu et al., 2024a; Feng et al., 2024; Shen et al., 2024b; Xie et al., 2024; Wan et al., 2025; Xia et al., 2025; Xia & Liu, 2025). While they all rely on slow per-scene optimization, we reconstruct fine details with a fast network forward pass. Closer to our setting is Zhang et al. (2025c), which proposes a unified frequency-aware radiance field to capture both normal-resolution global scene structure and tiny details in an area of interest, given high-resolution captures of the area. We share the same goal of leveraging local high-resolution observations, but pursue it in the context of generalizable 3D Gaussians.

## 3 METHOD

Given a set of 3D Gaussians $\mathcal{G}^{input}$ reconstructed from low-resolution input images, our goal is to densify and refine $\mathcal{G}^{input}$ to reconstruct high-resolution details in a user-specified local region of interest (RoI) based on additional high-resolution images of the specified region.

Concretely, we start with a sparse set of low-resolution input images $\mathcal{I}^{low}$ with known camera parameters, and apply state-of-the-art feed-forward 3D Gaussian reconstruction method Depth-Splat (Xu et al., 2024a) to obtain the initial 3DGS reconstruction $\mathcal{G}^{input}$.

We take another sparse set of high-resolution images capturing the RoI, denoted by $\mathcal{I} = \{I_i\}_{i=1}^N, I_i \in \mathbb{R}^{H \times W \times 3}$, with known camera projection matrices $\{\mathbf{P}_i\}_{i=1}^N$. The RoI is specified by its 2D projections onto the input views, given as binary masks $\mathcal{M} = \{M_i\}_{i=1}^N, M_i \in \{0,1\}^{H \times W}$.

To tackle the problem, we design a cross-modal Gaussian densification prediction framework *GaussianLens* (Sec. 3.1). It fuses multi-modal information from a set of initial Gaussians $\mathcal{G}^{init}$ and 2D images, and predicts a set of densified Gaussians $\mathcal{G}^{den}$. We divide the input Gaussians $\mathcal{G}^{input}$ into those in the RoI ($\mathcal{G}^{RoI}$) and the background ($\mathcal{G}^{bg}$), apply *GaussianLens* to $\mathcal{G}^{RoI}$ for local densification, i.e. $\mathcal{G}^{init} \leftarrow \mathcal{G}^{RoI}$, and merge the results with the background, i.e., $\mathcal{G}^{final} = \mathcal{G}^{den} \cup \mathcal{G}^{bg}$.

To better reconstruct details, we propose a novel pixel-guided Gaussian densification mechanism (Sec. 3.2) that augments $\mathcal{G}^{RoI}$ with another set of pixel-guided Gaussians $\mathcal{G}^{pixel}$, before applying *GaussianLens* to their union, i.e., $\mathcal{G}^{init} \leftarrow \mathcal{G}^{RoI} \cup \mathcal{G}^{pixel}$. The full workflow is shown in Fig. 2 (f).

### 3.1 *GaussianLens*: CROSS-MODAL GAUSSIAN DENSIFICATION PREDICTION FRAMEWORK

Given initial Gaussians $\mathcal{G}^{init}$ and 2D images, our Gaussian densification framework *Gaussian-Lens* (illustrated in Fig. 2) constructs initial multi-modal features, encodes them into latent features through a transformer network, which employs cross-attention to images to further fuse information, and finally decodes latent features into the parameters of new densified Gaussians $\mathcal{G}^{den}$. For brevity, below we present a simplified description. Please refer to Sec. B.1 for full details.

**Multi-Modal Residual Gaussian Feature Initialization** As shown in Fig. 2 (a), given input images $\mathcal{I}$ and initial Gaussians $\mathcal{G}^{init}$, we first construct initial per-Gaussian features $\{g_j\}$ and image features $\{H_i\}$ by fusing 3D Gaussian and 2D multi-view information.

To associate input 3D Gaussians with 2D images, we render $\mathcal{G}^{init}$ at input views and compare them with the input images. Concretely, at each view $i$, we obtain reconstructed RGB, depth, and opacities $(\hat{I}_i, \hat{D}_i, \hat{A}_i)$, and explicitly compute the residual between groundtruth images and reconstruction as $E_i = I_i - \hat{I}_i$. We define the reconstruction-residual image features as $H_i^{recon} = (\hat{I}_i, \hat{D}_i, \hat{A}_i, E_i)$. We also extract dense multi-view features $\{H_i^{\mathrm{mv}}\}_{i=1}^N$ with a pretrained multi-view feature extractor from Xu et al. (2023). Finally, we construct image features $\{H_i\}$ as $H_i = (I_i, H_i^{\mathrm{mv}}, H_i^{recon})$.

We construct initial Gaussian features $\{g_j\}$ as $g_j = (g_j^{param}, g_j^{grad}, g_j^{proj})$, where $g_j^{param}$ are the Gaussian parameters, $g_j^{grad}$ are the gradients of the rendering loss $\mathcal{L} = \sum_i ||E_i||^2$ with respect to the parameters, i.e., $\nabla_{G_j}\mathcal{L}$. We incorporate them to leverage the residual signal, similar to Chen et al. (2024b). $g_j^{proj}$ refers to local image features at the projected Gaussian centers. Concretely, we compute the 2D projection of Gaussian $G_j$'s center to view $i$, take the bilinear interpolation of image feature $H_i$ at the projection, and concatenate projection features from all views to obtain $g_j^{proj}$.

**PointTransformer-based Feature Encoder with Projection-Based Cross-Attention** We encode initial Gaussian features $\{g_j\}$ into latent features $\{f_j\}$ with an encoder network $\phi_{enc}$ based on PointTransformerv3 (PTv3) (Wu et al., 2024), as shown in Fig. 2 (b). We adopt its standard U-Net architecture with serialized self-attention layers, progressive down-/up-sampling and skip connections to efficiently extract spatial features at multiple scales.

We further propose a **projection-based cross-attention layer** (`ProjCrossAttn`) to fully integrate image information. As Fig. 2 (e) illustrates, given image features $\{H_i\}_{i=1}^N$ and projection matrices $\{\mathbf{P}_i\}_{i=1}^N$, for each Gaussian feature $f_j$ centered at $p_j$, `ProjCrossAttn` projects the 3D Gaussian center $p_j$ to each view $i$, bilinearly-interpolates image features $H_i$ at the 2D projection to obtain sampled image feature $h_{i,j} = H_i[\pi_{\mathbf{P}_i}(p_j)]$. Finally, it applies a standard cross-attention with Gaussian feature $f_j$ as the query and sampled image features $\{h_{i,j}\}_{i=1}^N$ as key and values. Formally,

$$\texttt{ProjCrossAttn}((p_j, f_j), \{H_i, \mathbf{P}_i\}_{i=1}^N) = \texttt{CrossAttn}(\{f_j\}, \{h_{i,j}\}_{i=1}^N), \quad h_{i,j} = H_i[\pi_{\mathbf{P}_i}(p_j)],$$

where `CrossAttn` is a standard pair-wise cross-attention between token sets $\{f_j\}$ and $\{h_{i,j}\}_{i=1}^N$. This projection-based approach establishes localized correspondences and is more scalable than global cross-attention that relates all Gaussians to all image tokens (Fan et al., 2024b).

To facilitate information exchange across varying scales, we build a multi-scale image feature pyramid $(\mathbf{H}, \mathbf{H}^{\frac{1}{2}}, \mathbf{H}^{\frac{1}{4}})$, and apply `ProjCrossAttn` to the last three decoder blocks of PTv3, fusing low-resolution image features with downsampled, low-resolution Gaussians and, conversely, high-resolution image features with full-resolution Gaussians.

In summary, $\phi_{enc}$ processes Gaussian features $\{(\mu_j, g_j)\}_{j=1}^M$ and image features $\mathbf{H}$ with spatial self-attention and multi-scale projection-based cross-attention to produce per-Gaussian features $\{f_j\}_{j=1}^M$.

**Gaussian Densification Decoder** As shown in Fig. 2 (c), for each initial Gaussian $G_j$, we use an MLP decoder $\phi_{dec}$ to map its final feature $f_j$ into the Gaussian parameters of $K$ final densified Gaussians $\{\hat{G}_j^k\}_{k=1}^K$, expressed as residuals to the original Gaussian parameters. Formally,

$$\hat{G}_j^k = G_j + \Delta\hat{G}_j^k, \quad \{\Delta\hat{G}_j^k\}_{k=1}^K = \{(\Delta\boldsymbol{\mu}_j^k, \Delta\alpha_j^k, \Delta\boldsymbol{\Sigma}_j^k, \Delta\boldsymbol{c}_j^k)\}_{k=1}^K = \phi_{dec}(f_j).$$

$K$ is a predefined densification factor, which we find sufficient to set to $K = 1$. [1] The final output of *GaussianLens* is the union of all densified Gaussians, i.e., $\mathcal{G}^{den} = \{\hat{G}_j^k\}_{j=1...M,k=1...K}$.

We train our model with mean squared error (MSE) between images rendered from predicted Gaussians and the groundtruth at $N'$ novel target test views $\{I_i\}_{i=1}^{N'}$, within RoI masks $\{M_i\}_{i=1}^{N'}$:

$$\mathcal{L}_{\text{MSE}} = \sum_{i=1}^{N'} \text{sum}(M_i \cdot ||I_i^{pred} - I_i^{GT}||_2^2) / \sum_{i=1}^{N'} \text{sum}(M_i)$$

### 3.2 PIXEL-GUIDED GAUSSIAN DENSIFICATION

Conceptually, *GaussianLens* densifies initial Gaussians $\mathcal{G}^{init}$ with learned cloning and refinement. However, for large resolution increases, solely cloning existing Gaussians is insufficient. For a $4\times$ zoom-in, a single initial Gaussian becomes responsible for capturing a $4 \times 4$-pixel region. Aggregating all information and mapping them to Gaussian parameters is a challenging learning task.

To address the challenge, we propose pixel-guided densification, a mechanism that directly injects information from the high-resolution observations by augmenting the initial Gaussians with a set of **pixel-guided Gaussians**, $\mathcal{G}^{\text{pixel}} = \{G_{i,xy}\}_{1 \leq i \leq N, M_{i,xy}=1}$. As shown in Fig. 2 (e), we spawn one new Gaussian $G_{i,xy}$ for each pixel $\mathbf{p}_{i,xy}$ within the RoI mask $M_i$ of each view $i$. The color $\mathbf{c}_{i,xy}$ of the Gaussian is initialized to the corresponding pixel color $I_{i,xy}$. The 3D position $\boldsymbol{\mu}_{i,xy}$ is determined by back-projecting the pixel along its camera ray to the depth rendered from the coarse initial 3DGS reconstruction. The opacity and scale are initialized to small constant values. Pixel-guided Gaussians explicitly incorporate dense appearance details, providing *GaussianLens* with a better foundation. We pass the union of input RoI Gaussians and pixel-guided Gaussians to *GaussianLens* for further densification and refinement, i.e., $\mathcal{G}^{init} \leftarrow \mathcal{G}^{RoI} \cup \mathcal{G}^{pixel}, \mathcal{G}^{den} = \text{GaussianLens}(\mathcal{G}^{init})$.

## 4 EXPERIMENT

### 4.1 REGION-OF-INTEREST VIEW SYNTHESIS BENCHMARK

**Dataset** We build our region-of-interest view synthesis benchmark based on RealEstate10K (RE10K) (Zhou et al., 2018) and DL3DV (Ling et al., 2024). RE10K contains real estate walk-through videos captured at $1280 \times 720$ resolution, while most feed-forward Gaussian works (Zhang et al., 2024c; Szymanowicz et al., 2024b; Chen et al., 2025b) use a downscaled $256 \times 256$ version, with few (Xu et al., 2024a) using higher resolution for qualitative results. DL3DV captures diverse, challenging scenes with $3840 \times 2160$ videos. So far, existing novel-view synthesis works (Ye et al., 2024a; Fischer et al., 2025; Ling et al., 2024; Seo et al., 2024; Chen et al., 2024a; Xu et al., 2024a; Kang et al., 2024) only use resolutions up to $960 \times 540$.

**Zoom-in Resolution Setting** On both datasets, we use $256 \times 256$ for low-resolution, full-sized input images for global capture. For high-resolution images focusing on the region-of-interest, we use $512 \times 512$ and $1024 \times 1024$ resolutions for RE10K and DL3DV, respectively. Given we only focus on local regions that constitute a small portion of the full-size image, to reduce redundancy and ease implementation, we only use $256 \times 256$ crops from high-res images that enclose the RoI projected masks. This can also be viewed as mimicking close-up captures with available high-res, global-scale images in the dataset. Below, we refer the two settings by "RE10K, $256 \rightarrow 512$ ($2\times$)" and "DL3DV, $256 \rightarrow 1024$ ($4\times$)" to emphasize the resolution increase and zoom-in factors.

**RoI Generation** To mimic the use case where users specify 3D RoIs via a 2D selection interface, we generate 3D RoIs by sampling 2D crops from context views and backprojecting them to the 3D scene. Please refer to Sec B.4 for more details.

**Evaluation Setting** We use the official train/test scene splits and follow the context/target view selection protocols of pixelSplat (Charatan et al., 2024) on RE10K and DepthSplat (Xu et al., 2024a) on DL3DV. We use the same views for both low-resolution and high-resolution images. We evaluate the results using standard novel view synthesis metrics PSNR, SSIM (Wang et al., 2004), and LPIPS (Zhang et al., 2018). We adapt them to the Region-of-Interest setting and only apply them

---

[1] To reduce redundancy and support selective densification, we can optionally predict an existence mask for each densified Gaussian. Please see Sec A.3 for more details.

to pixels within the RoI mask. We also report efficiency metrics, including the number of predicted Gaussians, trainable model parameters, and per-iteration training time and memory consumption.

## 4.2 BASELINES

We compare to the following adaptations of state-of-the-art feed-forward Gaussian reconstruction methods DepthSplat (Xu et al., 2024a), pixelSplat (Charatan et al., 2024), and MVSplat (Chen et al., 2025b). 1) **low-res full** is the standard model operating on $256 \times 256$ low-resolution, full-size inputs and predicts per-pixel Gaussians. We use this variant of DepthSplat to generate the initial Gaussians as inputs to our model. 2) **high-res full** uses high-resolution, full-size inputs ($512 \times 512$ on RE10K), and outputs per-pixel Gaussians at high-resolution. This variant has unfair access to additional information from the full-sized, high-resolution context images, and is therefore not directly comparable. 3) **high-res crop** takes $256 \times 256$ RoI crops from full-size high-resolution images as inputs, and only outputs Gaussians for pixels in the crop, avoiding the costly per-high-res-pixel prediction. Please refer to Sec. B.3 for details.

As all baselines predict new Gaussians of the RoI independent of the global initial ones, they do not produce a single, consistent reconstruction with both global content and local details. We only use them for reference in evaluating the reconstruction quality of the RoI.

## 4.3 BENCHMARK COMPARISONS

Table 1: **Quantitative comparisons on DL3DV and RE10K.** Best overall results are in **bold**. Best results under fair comparison, which excludes methods with privileged access to full high-res images (*), are underlined. Time and memory consumptions are measured with a batch size of 1 on an NVIDIA a6000 GPU.

| | Method | PSNR↑ | SSIM↑ | LPIPS↓ | Trainable Param.↓ | Training Cost Time↓ | Mem. ↓ | Num. GS↓ |
|---|---|---|---|---|---|---|---|---|
| RE10K $256 \to 512$ | pixelSplat low-res full | 25.94 | 0.820 | 0.128 | 118M | 0.89s | 14.32G | 393K |
| | MVSplat low-res full | 26.08 | 0.832 | 0.089 | 12M | 0.49s | 8.27G | 131K |
| | DepthSplat low-res full | 27.28 | 0.852 | 0.101 | 120M | 0.66s | 8.90G | 131K |
| | MVSplat high-res full* | 27.12 | 0.862 | 0.085 | 12M | 1.15s | 25.91G | 524K |
| | DepthSplat high-res full* | 28.18 | **0.876** | **0.083** | 120M | 2.03s | 32.66G | 524K |
| | DepthSplat high-res crop | 24.38 | 0.759 | 0.161 | 120M | 0.78s | 8.90G | 131K |
| | Ours | **28.46** | 0.874 | 0.087 | 43M | 1.74s | 13.27G | 214K |
| DL3DV $256 \to 1024$ | DepthSplat low-res full | 22.31 | 0.652 | 0.286 | 120M | 0.91s | 8.15G | 131K |
| | DepthSplat high-res full* | Out of Memory on an 80GB H100 GPU | | | | | | |
| | DepthSplat high-res crop | 19.51 | 0.571 | 0.352 | 120M | 0.87s | 8.10G | 131K |
| | Ours | **23.62** | **0.719** | **0.231** | 43M | 1.67s | 9.46G | 220K |

Table 1 shows the quantitative comparison with baselines. Our method consistently improves upon initial Gaussians predicted by 'DepthSplat low-res full' by more than 1dB PSNR, effectively leveraging high-resolution observations. Compared to 'Depthsplat high-res full' with unfair access to full-size high-resolution images, we achieve on-par performance on RE10K with only 40% Gaussian budgets, fewer model parameters, shorter training time, and much smaller memory consumption. Our method's efficiency and reduced computational requirements are more evident in the DL3DV $256 \to 1024$ setting, where 'DepthSplat high-res full' fails to train as it runs out of memory even on an 80GB H100 GPU, revealing the inefficiency and lack of scalability of standard, uniform-resolution feed-forward models. 'DepthSplat high-res crop' scales to high resolutions by operating on crops. However, it struggles to learn reliable multi-view geometry from the small-overlap, off-center local image crops, leading to lower performance. In contrast, our method effectively leverages initial Gaussians as a coarse 3D scaffold to aggregate information onto.

As shown in Fig. 3, our model improves details originally blurred out in low-resolution initial reconstructions, such as thin structures and fine patterns. Please see Sec. C for more qualitative results.

## 4.4 ZERO-SHOT GENERALIZATION TO DIFFERENT GAUSSIAN SOURCES

Despite trained exclusively on input 3D Gaussians predicted by DepthSplat, our method can zero-shot generalize to input 3D Gaussians from distinct sources: Gaussians predicted by unseen feed-

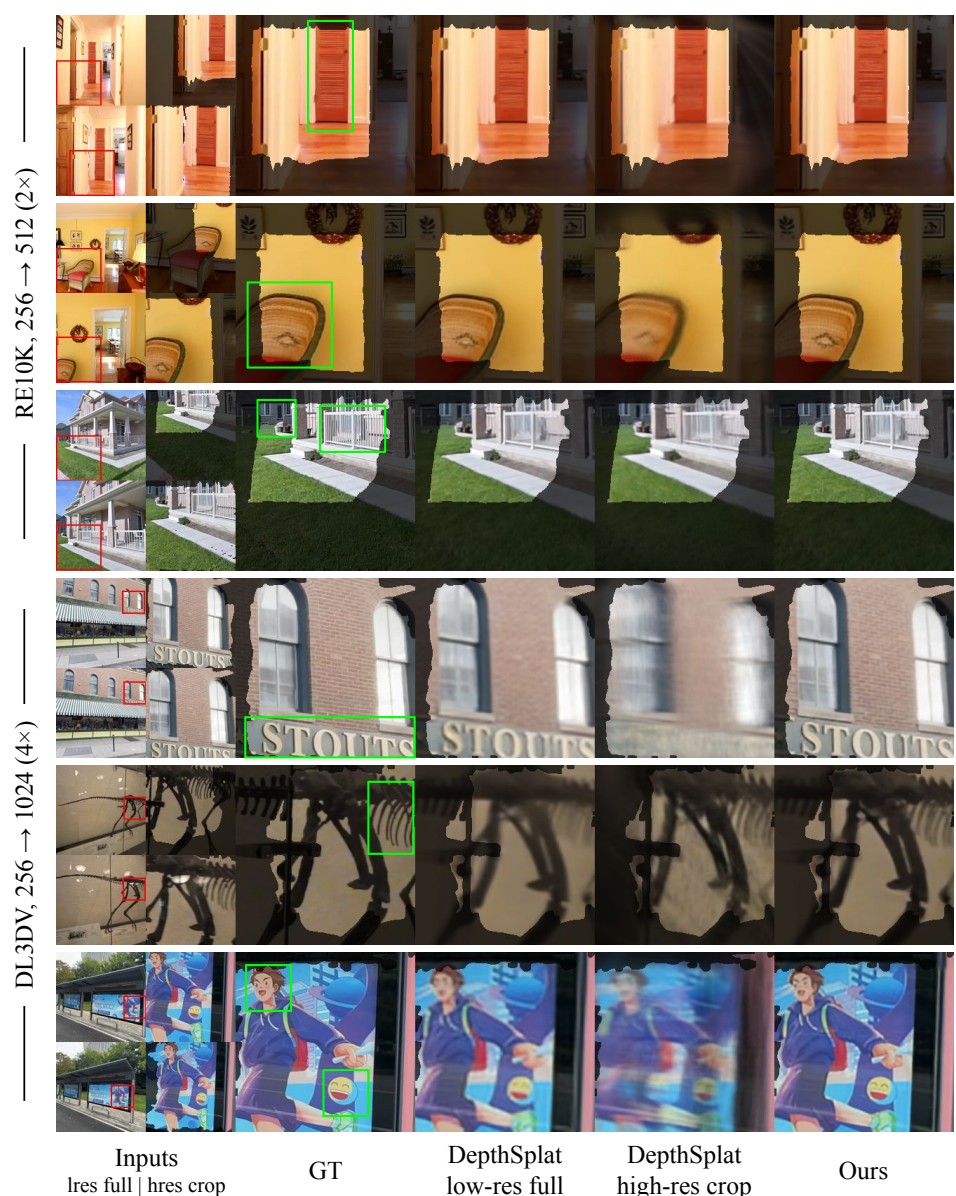

Figure 3: **Novel view synthesis on RealEstate10K (Zhou et al., 2018) and DL3DV (Ling et al., 2024).** Our method reconstructs finer details by effectively leveraging high-resolution observations and initial Gaussians.

forward models pixelSplat and MVSplat on RE10K, and Gaussians per-scene optimized from dense observations on DL3DV. As summarized in Table 2, our model achieves consistent improvement upon the input Gaussians in all metrics, showing strong generalization ability to inputs from different distributions. This is crucial for the flexible deployment of the method and underscores the advantage of not relying on prior knowledge or access to the internal features of the Gaussian source. Please refer to Sec A.1 for details and qualitative results.

## 4.5 ABLATION STUDIES

We ablate our method under the DL3DV $256 \rightarrow 1024$ setting and summarize the results in Table 3.

**Source Gaussians to Densify from.** Our method densifies Gaussians from both the input ($\mathcal{G}^{RoI}$) and pixel-guided densification ($\mathcal{G}^{pixel}$). As shown in Tab. 3a, densifying from either input Gaussians only ('input, $\kappa\times$'), or pixel-guided Gaussians only ('pixel') leads to decreased performance. Simply

Table 2: **Generalization to different input Gaussian sources**. We take our models trained exclusively on input Gaussians predicted by the DepthSplat low-res full model, and directly apply them to Gaussians from unseen feed-forward models pixelSplat and MVSplat on RE10K, or per-scene optimized Gaussians on DL3DV.

| Dataset | Source of input Gaussians | input Gaussians | | | predicted densification | | |
|---|---|---|---|---|---|---|---|
| | | PSNR↑ | SSIM↑ | LPIPS↓ | PSNR↑ | SSIM↑ | LPIPS↓ |
| RE10K $256 \rightarrow 512$ | DepthSplat low-res full | 27.28 | 0.852 | 0.101 | 28.46(+1.18) | 0.874(+0.022) | 0.087(-0.014) |
| | pixelSplat low-res full | 25.94 | 0.820 | 0.128 | 27.09(+1.15) | 0.844(+0.024) | 0.108(-0.020) |
| | MVSplat low-res full | 26.08 | 0.832 | 0.089 | 27.32(+1.24) | 0.853(+0.021) | 0.087(-0.002) |
| DL3DV $256 \rightarrow 1024$ | DepthSplat low-res full | 22.31 | 0.652 | 0.286 | 23.62(+1.31) | 0.719(+0.067) | 0.231(-0.055) |
| | per-scene optim. | 22.34 | 0.659 | 0.291 | 24.46(+2.12) | 0.742(+0.083) | 0.225(-0.066) |

Table 3: **Ablation studies.** All models are trained and evaluated under the DL3DV $256 \rightarrow 1024$ setting.

| GS Source | PSNR↑ | SSIM↑ | LPIPS↓ |
|---|---|---|---|
| input,$_{4\times}$ | $23.27_{+0.96}$ | 0.695 | 0.260 |
| input,$_{16\times}$ | $23.25_{+0.94}$ | 0.697 | 0.260 |
| pixel | $22.97_{+0.66}$ | 0.694 | 0.248 |
| Ours (both) | $23.62_{+1.31}$ | 0.719 | 0.231 |

| Init. Feat. | PSNR↑ | SSIM↑ | LPIPS↓ |
|---|---|---|---|
| no grad. | $23.46_{+1.16}$ | 0.711 | 0.239 |
| no recon. | $23.19_{+0.88}$ | 0.703 | 0.244 |
| no m.view | $23.41_{+1.10}$ | 0.708 | 0.241 |
| Ours | $23.62_{+1.31}$ | 0.719 | 0.231 |

| Attn. | PSNR↑ | SSIM↑ | LPIPS↓ |
|---|---|---|---|
| no attn. | $22.59_{+0.28}$ | 0.661 | 0.276 |
| global | $22.64_{+0.33}$ | 0.663 | 0.274 |
| last block | $23.47_{+1.17}$ | 0.714 | 0.236 |
| Ours | $23.62_{+1.31}$ | 0.719 | 0.231 |

(a) **Source Gaussians to densify from.** 'input, $K\times$' only densifies the original Gaussians $\mathcal{G}^{RoI}$ by densification factor $K$($K$= 4 or 16). 'pixel' only densifies pixel-guided Gaussians $\mathcal{G}^{pixel}$. Ours densifies both sources of Gaussians $\mathcal{G}^{RoI} \cup \mathcal{G}^{pixel}$.

(b) **Initial features.** 'no grad' removes gradient feature $g^{grad}$ from initial Gaussian features. 'no recon.', and 'no m.view' remove reconstruction features $H^{recon}$ and multi-view features $H^{mv}$ from initial image features.

(c) **Image-point cross-attention.** 'no attn.' removes cross-attention. 'global' performs attention between all images and points at the bottleneck. 'last block' only performs projection-based cross-attention at the last block.

increasing the densification factor $K$ from 4 to 16 does not lead to further improvement, implying the limitation of using input Gaussians only and the necessity of pixel-guided densification.

**Multi-Modal Residual Feature Initialization.** To analyze the construction of the initial Gaussian and image features, we remove gradient-based features $g^{grad}$ from initial Gaussian features ('no grad.'), reconstruction-residual features $H^{recon}$ or multi-view features $H^{mv}$ from image features ('no recon.' and 'no m.view'). As shown in Tab. 3b, all features contribute to the final performance. The most significant drop occurs when we remove image reconstructions and residuals rendered from initial Gaussians ('no recon.'), highlighting the importance of explicitly associating Gaussians and images through rendering for effective residual learning.

**Projection-Based Gaussian-Image Cross-attention.** We replace our multi-scale, projection-based cross-attention with 1) no cross-attention with images ('no attn.'), PTv3 performs self-attention only; 2) a global cross-attention between all image and Gaussian tokens ('global'), similar to Fan et al. (2024b), which is only applied to the bottleneck block with downsampled tokens due to the squared computation complexity; 3) a single-scale projection-based cross-attention at the last transformer block ('last block'). As shown in Tab. 3c, both no attention and global attention suffer significant performance drop, suggesting the vital role of local image information in Gaussian densification prediction. Multi-scale attention further improves performance upon single-scale ('last block').

## 5 CONCLUSION

In this paper, we introduce the task of localized high-resolution reconstruction via on-demand Gaussian densification. Our formulation addresses the practical need for spatially varying detail reconstruction, avoids the prohibitive cost of uniformly high-resolution reconstruction, and enables the effective use of high-resolution observations. We develop *GaussianLens*, a generalizable densification prediction framework that effectively fuses multi-modal input information. We further propose a novel pixel-guided densification mechanism to capture details under significant resolution increases. Our model achieves state-of-the-art performance in localized high-resolution reconstruction while consuming fewer computational resources.

**Limitations and Future Directions** Our model focuses on improving detail reconstruction based on a coarse initial Gaussian reconstruction. It is not designed to handle catastrophic errors already present in the initial reconstruction. Developing a method to leverage emerging geometry cues from high-resolution observations to recover from reconstruction failures at low resolution would be an exciting future direction.

## REPRODUCIBILITY STATEMENT

We are committed to ensuring the reproducibility of our work. To this end, we provide the model, implementation, and benchmark details in Appendix Sec. B. We will also release the source code and benchmark dataset upon acceptance.

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

## A    ADDITIONAL EXPERIMENTS AND ANALYSIS

### A.1    GENERALIZATION TO PER-SCENE OPTIMIZED 3DGS

In this section, we present additional details and qualitative results on zero-shot generalization to per-scene optimized 3DGS on DL3DV, which we introduced in Sec. 4.4.

We repurpose the per-scene optimized 3D Gaussian reconstructions of the 140 test scenes from DL3DV (Ling et al., 2024), which we used for sampling Regions-of-Interest (RoI) for the evaluation benchmark (detailed in Sec B.4). We initialize the 3D Gaussians with the Structure-from-Motion (Snavely et al., 2006) points from DL3DV's official COLMAP cache release, and optimize them with $480 \times 280$ (1/8) resolution images for 7K iterations. Consistent with the observation of the original 3DGS paper (Kerbl et al., 2023), the reconstructions at 7K iterations already capture the overall scene geometry and appearance. They are sufficient for our original purpose of sampling RoI, and serve as an interesting testbed for evaluating the generalization ability of our model.

Figure 4 shows qualitative examples of our generalization results. We compare the initial Gaussians and our corresponding densifications side-by-side for both DepthSplat predicted and per-scene optimized input Gaussians. As shown in the first two rows, our method improves detail reconstruction upon input Gaussians from both sources, demonstrating strong zero-shot generalization capability. The last two rows illustrate cases where per-scene optimization provides a better initialization, thanks to dense observations. In contrast, DepthSplat predictions suffer from the challenging sparse-view setting, leading to floaters in the third row and wrongly angled door structures in the fourth row. Our model is able to leverage the better starting point from per-scene optimization and produce more accurate final reconstructions.

### A.2    EXTENSION TO HIGHER RESOLUTIONS

We show additional results on two higher-resolution settings on DL3DV, where the original image resolution is $2160 \times 3840$: 1) $4\times$ zoom-in from $256 \times 480$ to $1024 \times 1920$, and 2) $8\times$ zoom-in from $256 \times 256$ to $2048 \times 2048$ (largest square from the original DL3DV images). Similar to the main experiments, we finetune the DepthSplat 'low-res, full' variants using low-resolution input images and high-resolution supervision, then train our model to refine and densify the 3DGS produced

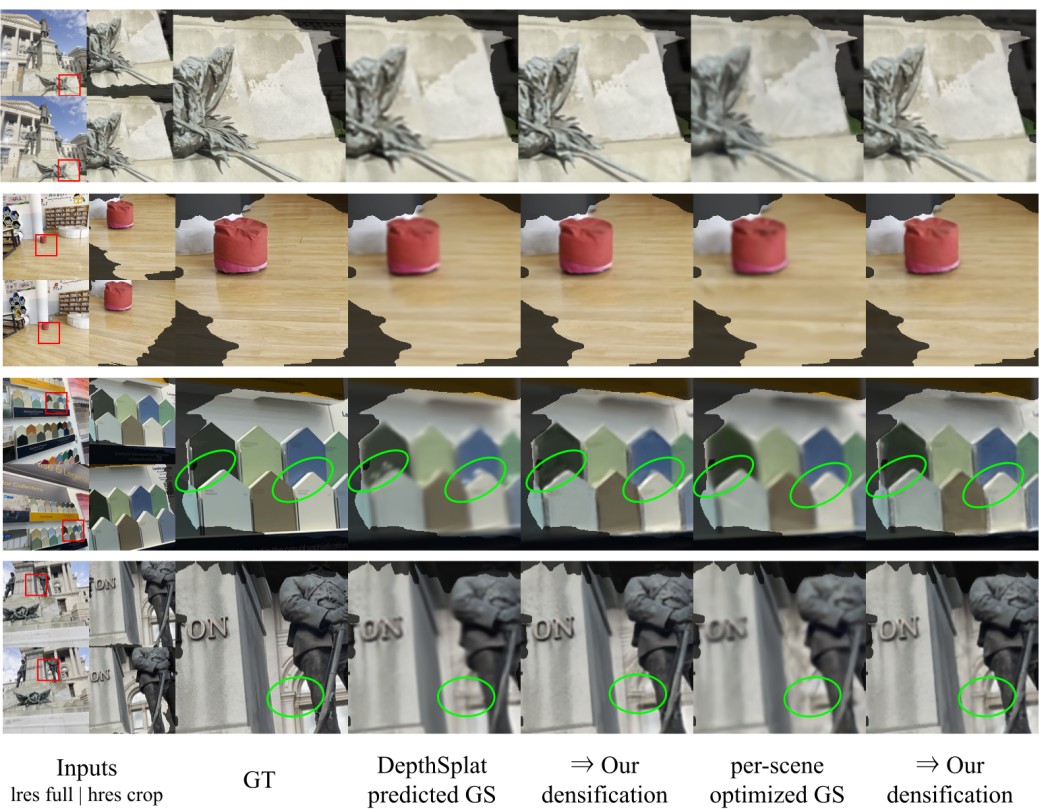

| Inputs | GT | DepthSplat | ⇒ Our | per-scene | ⇒ Our |
| lres full \| hres crop | | predicted GS | densification | optimized GS | densification |

Figure 4: **Our densification results given input Gaussians from DepthSplat prediction or per-scene optimization.** Our model achieves zero-shot generalization to per-scene optimized 3D Gaussians, improving upon initial reconstructions from both sources. The last two rows illustrate cases where per-scene optimization provides a more robust initialization, while DepthSplat struggles with sparse-view ambiguity, resulting in floaters in the third row, and wrongly angled door structure in the fourth row. Our model can leverage improved initial Gaussians and produce more accurate final reconstructions.

by finetuned DepthSplat. However, due to limited storage space, we only trained our models on DL3DV-3K and DL3DV-4K splits, a 2K-scene subset of the complete DL3DV dataset. We perform the evaluation on the 140 standard DL3DV test scenes.

Table 4: **Extension to higher resolutions**. We experiment with two additional higher-resolution settings on DL3DV, where our method consistently improves the initial Gaussian reconstruction from DepthSplat.

| Resolution Setting | DepthSplat low-res full | | | predicted densification | | |
|---|---|---|---|---|---|---|
| | PSNR↑ | SSIM↑ | LPIPS↓ | PSNR↑ | SSIM↑ | LPIPS↓ |
| $256 \times 480 \rightarrow 1024 \times 1920$ (4×) | 22.76 | 0.685 | 0.283 | 23.98(+1.22) | 0.739(+0.054) | 0.233(-0.050) |
| $256 \times 256 \rightarrow 2048 \times 2048$ (8×) | 22.32 | 0.653 | 0.330 | 23.24(+0.92) | 0.692(+0.039) | 0.281(-0.049) |

As shown in Table 4, our method consistently improves the input 3DGS from DepthSplat at higher-resolution settings, including the challenging case of 8× zoom-in.

## A.3 LEARNING EXISTENCE MASKS FOR SELECTIVE DENSIFICATION

To reduce redundancy and support selective densification, conceptually, our framework can additionally predict an existence probability $p_j^k \in [0, 1]$ for each densified Gaussian $G_j^k$. To reduce

Table 5: **Selective densification by learning existence masks**. We perform the experiments under the DL3DV $256 \rightarrow 1024$ setting. Training our model with a mask regularization loss achieves performance close to the full model with $80\%$ of Gaussians, suggesting the potential to further reduce the number of Gaussians and the flexibility to balance quality and cost.

| Method | $w_{mask}$ | PSNR↑ | SSIM↑ | LPIPS↓ | Num. GS↓ |
|--------|-----------|-------|-------|--------|----------|
| Ours | 0 | 23.62 | 0.719 | 0.231 | 220K |
| masked | 0.0001 | 23.56 | 0.711 | 0.239 | 170K |

the final number of Gaussians, we can apply a regularization loss $\mathcal{L}_{mask}$ to $\{p_j^k\}$, and only keep Gaussians with high existence probabilities.

Implementation-wise, we adapt MaskGaussian (Liu et al., 2024b), a per-scene 3D Gaussian optimization method. For Gaussian $G_j$ ($k$ omitted for simplicity), the decoder predicts two mask logits $m_j^Y, m_j^N$, corresponding to the categories "exists" and "does not exist". A discrete yet differentiable category $M_j \in \{0, 1\}$ is then sampled with Gumbel Softmax (Jang et al., 2016), determining whether $G_j$ is active in the current pass. The Gaussian parameters and masks $\{(G_j, M_j)\}$ are then processed by MaskGaussian's specialized rasterizer, which renders active Gaussians with $M_j = 1$ during the forward pass. In the backward pass, it computes gradients both with respect to the parameters of rendered Gaussians, and the mask values $M_j$ of all Gaussians. The gradients are backpropagated all the way to the mask logit decoder, enabling learnable masking and selective densification.

The mask regularization loss is formally defined as:

$$\mathcal{L}_{mask} = \text{mean}(\sum_{j,k} M_j^k + \sum_{i,xy,k} M_{i,xy}^k)^2,$$

where the two terms account for Gaussians densified from input RoI Gaussians $\mathcal{G}^{RoI}$ and pixel-guided Gaussians $\mathcal{G}^{pixel}$, respectively.

The final objective is $\mathcal{L} = \mathcal{L}_{\text{MSE}} + w_{mask}\mathcal{L}_{mask}$, with $w_{mask} = 0$ in main experiments to prioritize reconstruction quality.

As shown in Table 5, the variant trained with mask regularization ($w_{mask} = 0.0001$) achieves performance close to the full model while using only $80\%$ Gaussians, indicating our framework's potential to flexibly trade off reconstruction quality and computational cost.

## A.4 VISUALIZING THE ROLES OF INPUT GAUSSIANS AND PIXEL-GUIDED DENSIFICATION

For an intuitive understanding of the contribution from input Gaussians and pixel-guided densification, we show a breakdown visualization of source and predicted Gaussians in Figure 5. The network learns to update both input coarse Gaussians $\mathcal{G}^{RoI}$ and pixel-guided Gaussians $\mathcal{G}^{pixel}$ to collaboratively reconstruct the scene. While some details emerge from the input Gaussians, they serve more as a coarse backdrop, on which pixel-guided Gaussians render sharp details.

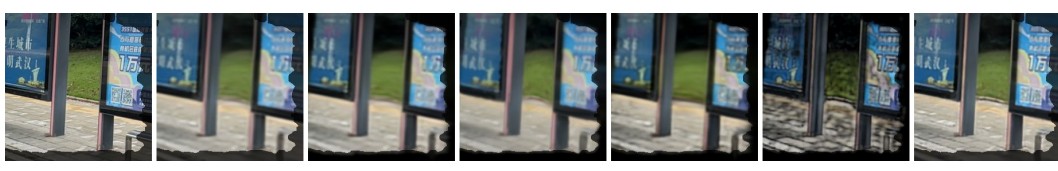

(a) GT     (b) input GS     (c) input GS in RoI     (d) pixel-guided GS     (e) updated input GS     (f) updated pixel GS     (g) output GS

Figure 5: **Breakdown visualization of source and output Gaussians**. Starting from input Gaussians (b), our network takes input Gaussians in the specified RoI (c), and Gaussians from pixel-guided densification (d), and outputs updated versions of both (e, f). While the emergence of more details can already be observed in updated input Gaussians (e), Gaussians from pixel-guided densification (f) reconstruct sharp details more effectively.

# B METHOD DETAILS

## B.1 MODEL DETAILS

**Feature Initialization** To effectively associate input 3D Gaussians and 2D images, we render the 3D Gaussians at input views and compare the results with the input images. Note that we render all Gaussians $\mathcal{G}^{input}$ for the complete reconstruction. Concretely, we obtain reconstructed RGB, depth, and accumulated opacities at view $i$ as

$$(\hat{I}_i, \hat{D}_i, \hat{A}_i) = \text{Rasterize}(\mathcal{G}^{input}; \mathbf{P}_i),$$

where $\mathbf{P}_i$ is the projection matrix of view $i$. We also explicitly compute the residual between groundtruth images and reconstruction as $E_i = I_i - \hat{I}_i$. We define the reconstruction-residual image features as

$$H_i^{recon} = (\hat{I}_i, \hat{D}_i, \hat{A}_i, E_i).$$

While $H_i^{recon}$ accounts for the totality of the input reconstruction $\mathcal{G}^{input}$, we only densify the subset $\mathcal{G}^{RoI}$ and keep the rest $\mathcal{G}^{bg}$ the same. Therefore, we also render the unchanged Gaussians $\mathcal{G}^{bg}$ alone to encode background or context information. Specifically, we compute

$$H_i^{bg\_recon} = (\hat{I}_i^{bg}, \hat{D}_i^{bg}, \hat{A}_i^{bg}) = \text{Rasterize}(\mathcal{G}^{bg}; \mathbf{P}_i)$$

The image features $\{H_i\}$ are constructed as

$$H_i = \text{RayModulate}(I_i, H_i^{\text{mv}}, H_i^{recon}, H_i^{bg\_recon}; \mathbf{P}_i),$$

where $H_i^{\text{mv}} \in \mathbb{R}^{H \times W \times C}$ is a multi-view feature from an off-the-shelf image encoder (Xu et al., 2023; 2024a). To further exploit our knowledge of the camera poses, we use $\text{RayModulate}(\cdot; \mathbf{P}_i)$ to incorporate per-pixel camera ray information, following Chen et al. (2025a). We compute the Plücker coordinates for the camera rays at each pixel and use them to modulate the image feature map via adaptive layer norm (Peebles & Xie, 2023).

We construct initial Gaussian features $\{g_j\}$ as

$$g_j = (g_j^{param}, g_j^{grad}, g_j^{proj}),$$

where $g_j^{param}$ and $g_j^{grad}$ are the Gaussian parameters and gradients of the rendering loss $\mathcal{L} = \sum_i ||E_i||^2$ with respect to the parameters, i.e., $\nabla_{G_j}\mathcal{L}$. In particular, we represent the Gaussian parameters as $(\boldsymbol{\mu}_j, \alpha_j, \boldsymbol{s}_j, \boldsymbol{q}_j, \boldsymbol{c}_j)$, where we decompose the covariance matrix $\boldsymbol{\Sigma}_j \in \mathbb{R}^{3 \times 3}$ into scale $\boldsymbol{s}_j \in \mathbb{R}^3$ and rotation $\boldsymbol{q}_j \in \mathbb{R}^4$, represented as a quaternion.

$g_j^{proj}$ refers to features taken from images by projecting the Gaussian center to images. Concretely, for view $i$, we compute the 2D projection of Gaussian $G_j$'s center to view $i$, $\pi_{\mathbf{P}_i}(\boldsymbol{\mu}_j) \in \mathbb{R}^2$, and take the bilinear interpolation of image feature $H_i$ at the projection, denoted as $H_i[\pi_{\mathbf{P}_i}(\boldsymbol{\mu}_j)]$. We concatenate projection features from all views to obtain

$$g_j^{proj} = \text{concat}_{i=1}^N H_i[\pi_{\mathbf{P}_i}(\boldsymbol{\mu}_j)].$$

**Projection-Based Cross-Attention Layer** We construct the cross-attention layer following the structure of PointTransformerv3 (Wu et al., 2024)'s point serialized attention block. It consists of a residual cross-attention unit, with the point feature $f_j$ as queries and the image features from all views $\{H_i[\pi_{\mathbf{P}_i}(p_j)]\}_{i=1}^N$ as keys and values, and a residual MLP block. Layer normalization is applied to all features before applying attention or MLP.

**Densification Decoder** Given output Gaussian feature $f_j$ from the transformer backbone, we use Gaussian densification decoder $\phi_{dec}$ to map it to the Gaussian parameters of $K$ final densified Gaussians $\{\hat{G}_j^k\}_{k=1}^K$. To encourage diversification of the $K$ densified Gaussians, we first decode the position offsets $\{\Delta\boldsymbol{\mu}_j^k\}_{k=1}^K$ from $f_j$ with a two-layer MLP, then decode the other parameter offsets conditioned on both $f_j$ and $\Delta\boldsymbol{\mu}_j^k$ with another two-layer MLP. We also perform a final round of cross-attention with image features $H$ using the predicted Gaussian centers $\hat{\boldsymbol{\mu}}_j^k = \boldsymbol{\mu}_j + \Delta\boldsymbol{\mu}_j^k$, which provide a more accurate spatial association with local image features.

Similar to how the opacities and scales of cloned Gaussians are set to smaller values in the original Adaptive Density Control (Kerbl et al., 2023), we compute "default post-densification" Gaussian

parameters $\tilde{G}_j^{(K)}$ and predict offsets from them. We keep using the center, rotation, and color of the original Gaussian as default for the densified Gaussians, i.e., $\tilde{\boldsymbol{\mu}}_j^{(K)} = \boldsymbol{\mu}_j, \tilde{\boldsymbol{q}}_j^{(K)} = \boldsymbol{q}_j, \tilde{\boldsymbol{c}}_j^{(K)} = \boldsymbol{c}_j$, and follow Kheradmand et al. (2024) for the computation of opacity $\tilde{\alpha}_j^{(K)}$ and scale $\tilde{\boldsymbol{s}}_j^{(K)}$ based on the densification factor $K$. Intuitively, when predicted offsets are initialized as $\mathbf{0}$, $K$ new Gaussians with these default parameters provide a better approximation to the original Gaussian $G_j$.

**Region-of-Interest Gaussian Selection**   Given input 3D Gaussians $\mathcal{G}^{input}$ that reconstructs the scene globally, we focus on a local set of Gaussians $\mathcal{G}^{RoI}$ that reconstruct the Region of Interest (RoI), and keep the other Gaussians (referred to as background Gaussians $\mathcal{G}^{bg}$) intact. To select $\mathcal{G}^{RoI}$, we compute a binary mask $M_j^{RoI}$ over all Gaussians $G_j$. Given binary RoI masks at input views $\mathcal{M} = \{M_i\}_{i=1}^N$, we render $\mathcal{G}^{input}$ at each view $i$, compute each Gaussian $G_j$'s contribution $\text{contrib}_j^{(i)}$ to the total opacity in the masked region $M_i$, and consider $G_j$ as visible from view $i$ by thresholding $\text{contrib}_j^{(i)}$, i.e., $\text{visible}_j^{(i)} = [\text{contrib}_j^{(i)} > \tau]$, where we set $\tau = 0.1$. We consider initial Gaussians visible from at least two context views, i.e., $M_j^{RoI} = [\sum_{i=1}^N \text{visible}_j^{(i)} \geq 2]$, assuming that they are less likely to contain errors from single-view ambiguity and hallucination.

**Pixel-Guided Densification**   For each input view $i$, we consider all pixels $\mathbf{p}_{i,xy}$ with image coordinate $(x, y)$ within the RoI mask $M_i$, i.e. $M_i(x, y) = 1$, and create a Gaussian
$$G_{i,xy} = (\boldsymbol{\mu}_{i,xy}, \alpha_{i,xy}, \boldsymbol{\Sigma}_{i,xy}, \boldsymbol{c}_{i,xy}).$$
We set

$$\boldsymbol{\mu}_{i,xy} = \mathbf{o}_i + \hat{\text{depth}}_{i,xy} \cdot \mathbf{d}_{i,xy},$$
$$\alpha_{i,xy} = \alpha_{init},$$
$$\boldsymbol{\Sigma}_{i,xy} = s_{init} I_{3\times3},$$
$$\boldsymbol{c}_{i,xy} = I_i(x, y),$$

where $\mathbf{o}_i$ is the camera origin, $\mathbf{d}_{i,xy}$ is the ray direction vector corresponding to $(x, y)$ computed from camera projection matrix $\mathbf{P}_i$, $\hat{\text{depth}}_{i,xy}$ is the reconstructed depth at $(x, y)$, obtained by rasterizing $\mathcal{G}^{input}$. $\alpha_{init} = 0.05, s_{init} = 0.02$ are hyperparameters. $I_i(x, y)$ is the color at $(x, y)$.

We refer to the set of Gaussians created from pixels as $\mathcal{G}^{pixel}$. To better handle $\mathcal{G}^{pixel}$ and $\mathcal{G}^{RoI}$ that follow two different distributions, we adjust the feature initialization and network as follows.

For feature initialization, we can compute the Gaussian parameter features $g_{i,xy}^{param}$ and image-projection features $g_{i,xy}^{proj}$ for $G_{i,xy} \in \mathcal{G}^{pixel}$ as before. However, the rendering-based gradient features $g^{grad}$ and reconstruction-residual image features $H^{recon}$ require a holistic association between the images and the entire set of Gaussians. To accommodate this, we render the union of all Gaussians, $\mathcal{G}^{input} \cup \mathcal{G}^{pixel}$, and compute an additional set of gradient features $g^{grad+}$ and reconstruction-residual image features $H_{recon}^+$. Note that $g^{grad+}$ is computed for both $\mathcal{G}^{RoI}$ and $\mathcal{G}^{pixel}$. We still keep the original $g_j^{grad}$ for $G_j \in \mathcal{G}^{RoI}$ and set $g_{i,xy}^{grad} = \mathbf{0}$ for $G_{i,xy} \in \mathcal{G}^{pixel}$.

To sum up, the final initial Gaussian features and image features are computed as
$$g = (g^{param}, g^{grad}, g^{grad+}, g^{proj}),$$
$$H = \text{RayModulate}(I, H^{mv}, H^{recon}, H^{recon+}, H^{bg\_recon} ; \mathbf{P}).$$

For our transformer backbone, we attach learnable $d$-dim embeddings $e_{RoI}, e_{pixel}$ to input Gaussian features from the two sources, respectively. In the densification decoder, we also attach different learnable embeddings to features of different Gaussians, and learn two separate decoder heads to predict the final Gaussian parameters. Intuitively, one would focus on predicting residual updates of $\mathcal{G}^{RoI}$ conditioned on initial Gaussian parameters, while the other predicts some attributes, e.g. scales and opacities, as absolute values due to our uniform initialization of $\mathcal{G}^{pixel}$.

**Gaussian Set Operation Workflow**   We start by dividing input Gaussians into those within the RoI and those out of the RoI, i.e. $\mathcal{G}^{input} = \mathcal{G}^{RoI} \cup \mathcal{G}^{bg}$. We then introduce another set of Gaussians $\mathcal{G}^{pixel}$ by pixel-guided densification, and apply the densification framework to the union $\mathcal{G}^{RoI} \cup \mathcal{G}^{pixel}$ to

obtain a densified set of Gaussians $\mathcal{G}^{den} = \text{GaussianLens}(\mathcal{G}^{RoI} \cup \mathcal{G}^{pixel})$. Finally, we merge them with the out-of-RoI Gaussians to obtain the final refined reconstruction $\mathcal{G}^{final} = \mathcal{G}^{den} \cup \mathcal{G}^{bg}$.

## B.2 IMPLEMENTATION DETAILS

**Training** We implement our method in PyTorch (Paszke, 2019) and use an AdamW (Loshchilov, 2017) optimizer with a cosine learning rate schedule. We use a learning rate of $1 \times 10^{-4}$ and a weight decay factor of $0.01$. We train our model for 200K iterations with a batch size of 6 on RealEstate10K, and 200K iterations with a batch size of 4 on DL3DV.

**Rasterization** We implement our differentiable 3D Gaussian rasterizer based on MaskGaussian (Liu et al., 2024b), with important changes to account for

- **Anti-aliasing**. We adapt the $\sigma$ hyperparameter in the heuristic 2D dilation process (pointed out by Yu et al. (2024b)) to the rendering resolution, similar to the 2D scale-adaptive filter proposed in Song et al. (2024). This removes the immediate artifacts when we render the same set of Gaussians at a resolution different from training. While it is not a critical concern, as we use the same resolution for finetuning and evaluation, anti-aliasing facilitates finetuning from pretrained checkpoints that were trained with low-resolution supervision.

- **Median depth rendering**. We follow Luiten et al. (2024) and render per-pixel depth as the depth of the Gaussian center, which causes the accumulated rays transmittance to drop below $0.5$.

**Model** For feature initialization, we use a ViT-B monocular backbone and a 2-scale multi-view branch for the multi-view image encoder (Xu et al., 2023; 2024a). We freeze the model weights from Xu et al. (2024a), and only finetune the weights of a DPT head (Ranftl et al., 2021) attached to the encoder. For modulation with ray plücker coordinates, we follow the implementation from Chen et al. (2025a). For the transformer backbone, we follow the default PointTransformerv3 (Wu et al., 2024) architecture with 5 encoder blocks and 4 decoder blocks. We add cross-attention layers at the end of the last 3 decoder blocks, each uses an 8-head attention. The corresponding image pyramid features are downscaled to $1/4, 1/2, 1$, with $128, 96, 64$ channels, respectively.

## B.3 BASELINE DETAILS

For experiments on the RealEstate10K (Zhou et al., 2018) (RE10K) dataset, we finetune DepthSplat-based baselines from the official model checkpoint of the "Base" model (with a ViT-B monocular branch and a 2-scale multi-view branch) trained on 2-view, $256 \times 256$ resolution images. We finetune pixelSplat and MVSplat from their official checkpoints, both also trained on 2-view, $256 \times 256$ images. For experiments on the DL3DV (Ling et al., 2024) dataset, we finetune DepthSplat from the official model checkpoint trained on 2-view, $256 \times 448$ RealEstate10K dataset and finetuned on 2-6 views, $256 \times 448$ DL3DV (Ling et al., 2024) dataset. We finetune all baselines for 200K iterations.

Below, we detail each baseline variant using the RE10K, $256 \rightarrow 512$ setting as an example.

- 'low-res full' variants take two low-resolution ($256 \times 256$), full-sized input images, and generate $K$ Gaussians per input pixel ($K \times 2 \times 256 \times 256$ Gaussians in total, $K = 3$ for pixelSplat, $K = 1$ for MVSplat and DepthSplat, following their original settings). The Gaussians are rendered at high resolution ($512 \times 512$), and supervised with high-resolution ($512 \times 512$) groundtruth images.

- 'high-res full' variants take two high-resolution ($512 \times 512$), full-sized input images, and generate $K$ Gaussians per input pixel ($K \times 2 \times 512 \times 512$ Gaussians in total, $K = 1$ for MVSplat and DepthSplat). The Gaussians are rendered at high resolution ($512 \times 512$), and supervised with high-resolution ($512 \times 512$) groundtruth images.

- 'high-res crop' variants take $256 \times 256$ crops from two high-resolution ($512 \times 512$) input images that enclose the region of interest, and generate $K$ Gaussians per input pixel ($K \times 2 \times 256 \times 256$ Gaussians in total, $K = 1$ for DepthSplat). The Gaussians are rendered at high resolution (crops from $512 \times 512$), and supervised with high-resolution groundtruth images in the region of interest.

  As this setting involves resizing and non-centered cropping of images, we make our best effort to ensure the camera parameters are correctly modified. We also modify the Gaussian rasterizer

to support non-centered intrinsic matrices and partial images. The modifications are carefully verified, and the code will be released.

Meanwhile, we have confirmed with the authors that DepthSplat does not assume centered intrinsics and is compatible with the cropped setting.

### B.4 REGION OF INTEREST GENERATION

Given a 3D scene $\mathcal{S}$ and a set of views $i = 1, \ldots, N$, described with images and camera projection matrices $\{I_i, \mathbf{P}_i\}_{i=1}^{N}, I_i \in \mathbb{R}^{H \times W \times 3}, \mathbf{P}_i \in \mathbb{R}^{4 \times 4}$, we aim to generate a local 3D region of interest (RoI) $\mathcal{R}$ suitable for our local high-resolution reconstruction task, and compute its 2D projections at the views $i = 1, \ldots, N$, described as binary masks $\{M_i\}_{i=1}^{N}, M_i \in \mathbb{R}^{H \times W}$.

Consistent with real-world usage where selections are made via a 2D interface, we first sample a 2D region $R_i^{2D}$ from a random view $i$, which we set to $i = 1$ for simplicity. The 2D region $R_i^{2D}$ is then back-projected to the 3D scene $\mathcal{S}$ to obtain 3D region $\mathcal{R}$.

To constrain the selection to be local, we sample a fixed-sized $H_{crop} \times W_{crop}$ rectangle $R_i^{crop}$, where $H_{crop} = c \cdot H, W_{crop} = c \cdot W$. We use $c = 0.5$ for RE10K and $c = 0.25$ for DL3DV.

For back-projection to 3D, as groundtruth 3D scene $\mathcal{S}$ is typically not available, we use a 3D Gaussian reconstruction $\mathcal{G}$ as a proxy. To obtain $\mathcal{G}$, we run per-scene optimization using all available images on the 140 test scenes of DL3DV. Sec. A.1 provides more details. However, per-scene optimization takes more than 7 min even for a coarse reconstruction. Given limited resources, for the 7K test scenes of RE10K and even more training scenes in both datasets, we use DepthSplat to reconstruct $\mathcal{G}$ from two views. The back-projection process from 2D region to 3D Gaussians is the same as Region-of-Interest Gaussian selection in our method, except for only considering one view, please refer to Sec. B.1 for more details.

The backprojection process results in a set of Gaussians $\mathcal{G}^{R_{init}}$ that constitute the initial 3D RoI $\mathcal{R}_{init}$, which we will further prune and refine. We render $\mathcal{G}^{R_{init}}$ to all views and obtain binary masks $\{M_i^{init}\}$ by thresholding accumulated opacity $A_i^{init}$. For each mask $M_i^{init}$, we sample rectangular crop $R_i^{crop}$ of size $H_{crop} \times W_{crop}$, with higher probability given to crops that enclose more masked areas. We take the intersection of crops and masks as updated per-view binary RoI masks $M_i^{upd} = R_i^{crop} \cap M_i^{init}$. Finally, we re-compute Gaussians visible from at least two views masked by $M_i^{upd}$, denoted by $\mathcal{G}^{R_{final}}$, as our final 3D RoI. We render $\mathcal{G}^{R_{final}}$ to views $i = 1, \ldots, N$ for the final binary RoI masks $\{M_i\}$. We limit our selection to Gaussians or 3D regions visible from more than one view to avoid distraction from single-view ambiguity, and focus on the core of the problem, i.e., achieving better detail reconstruction.

## C ADDITIONAL QUALITATIVE RESULTS

We show more novel view synthesis comparisons on RealEstate10K and DL3DV between initial Gaussians predicted by DepthSplat and our predicted densification in Figure 6 7 8 9 and Figure 10 11 12 13.

## D LLM USAGE

We used LLM to aid and polish writing, including word choice and sentence rephrasing.

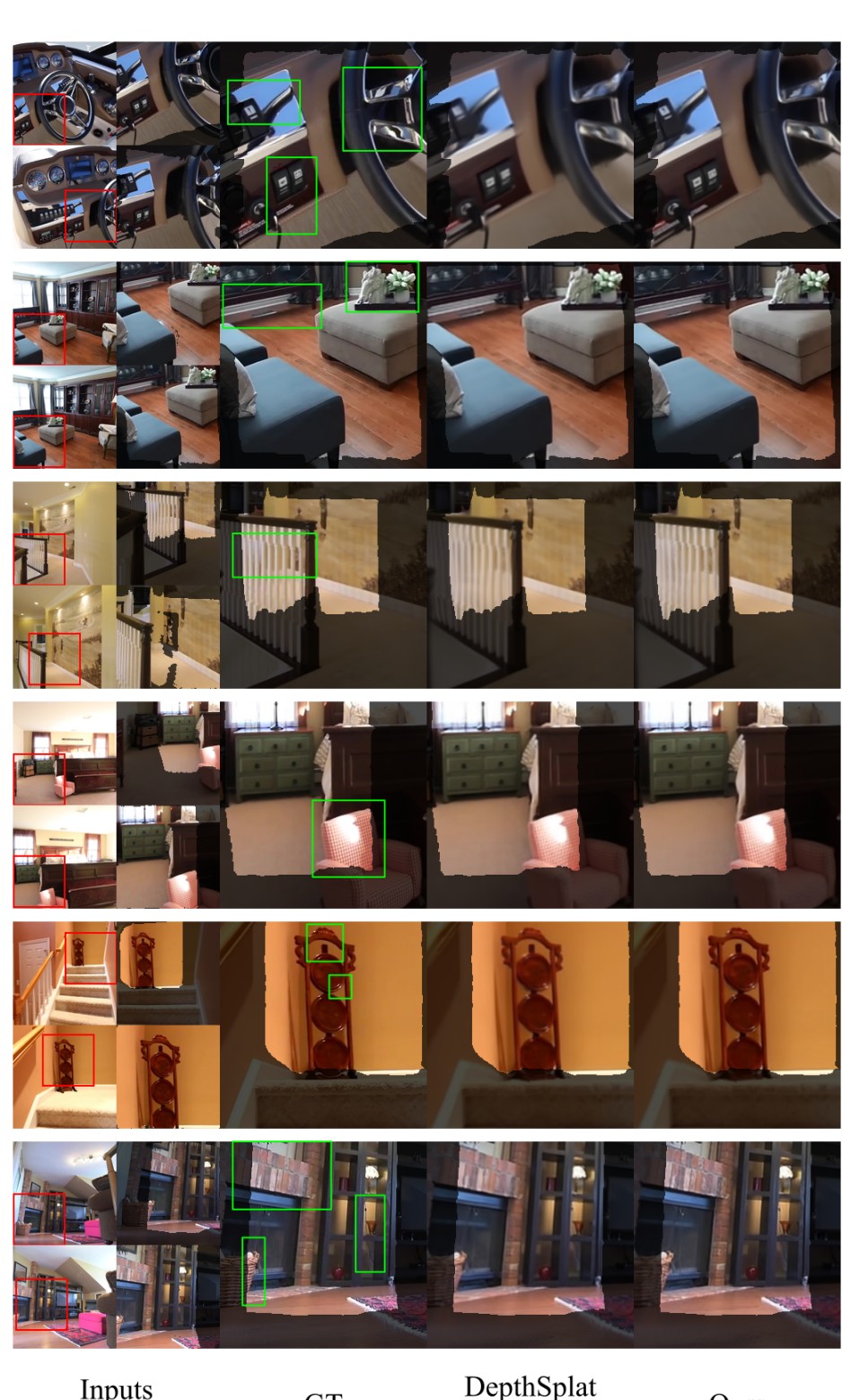

| Inputs | GT | DepthSplat | Ours |
| lres full \| hres crop | | low-res full | |

Figure 6: **Novel view synthesis on RealEstate10K (Zhou et al., 2018).**

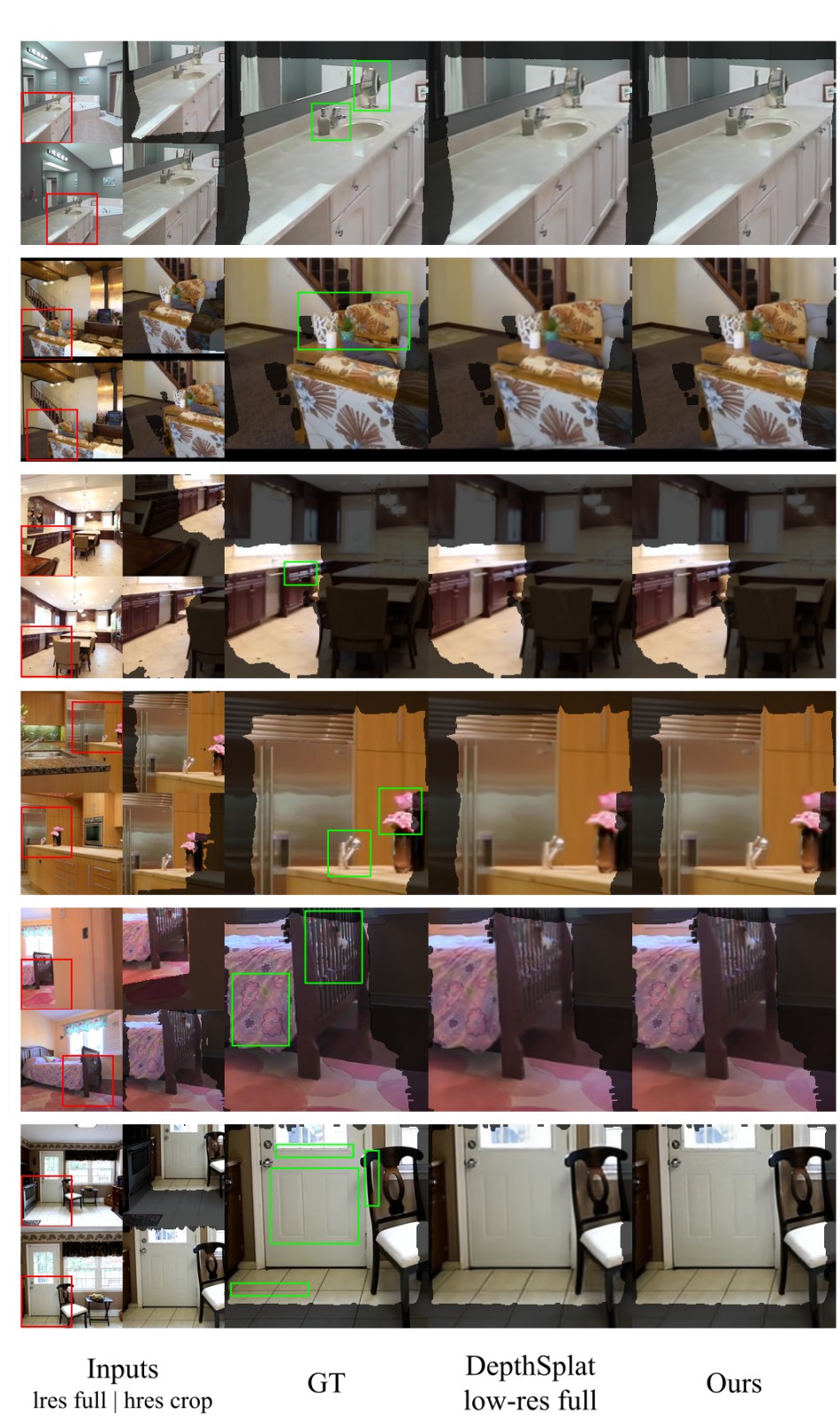

Inputs
lres full | hres crop

GT

DepthSplat
low-res full

Ours

Figure 7: **Novel view synthesis on RealEstate10K (Zhou et al., 2018).**

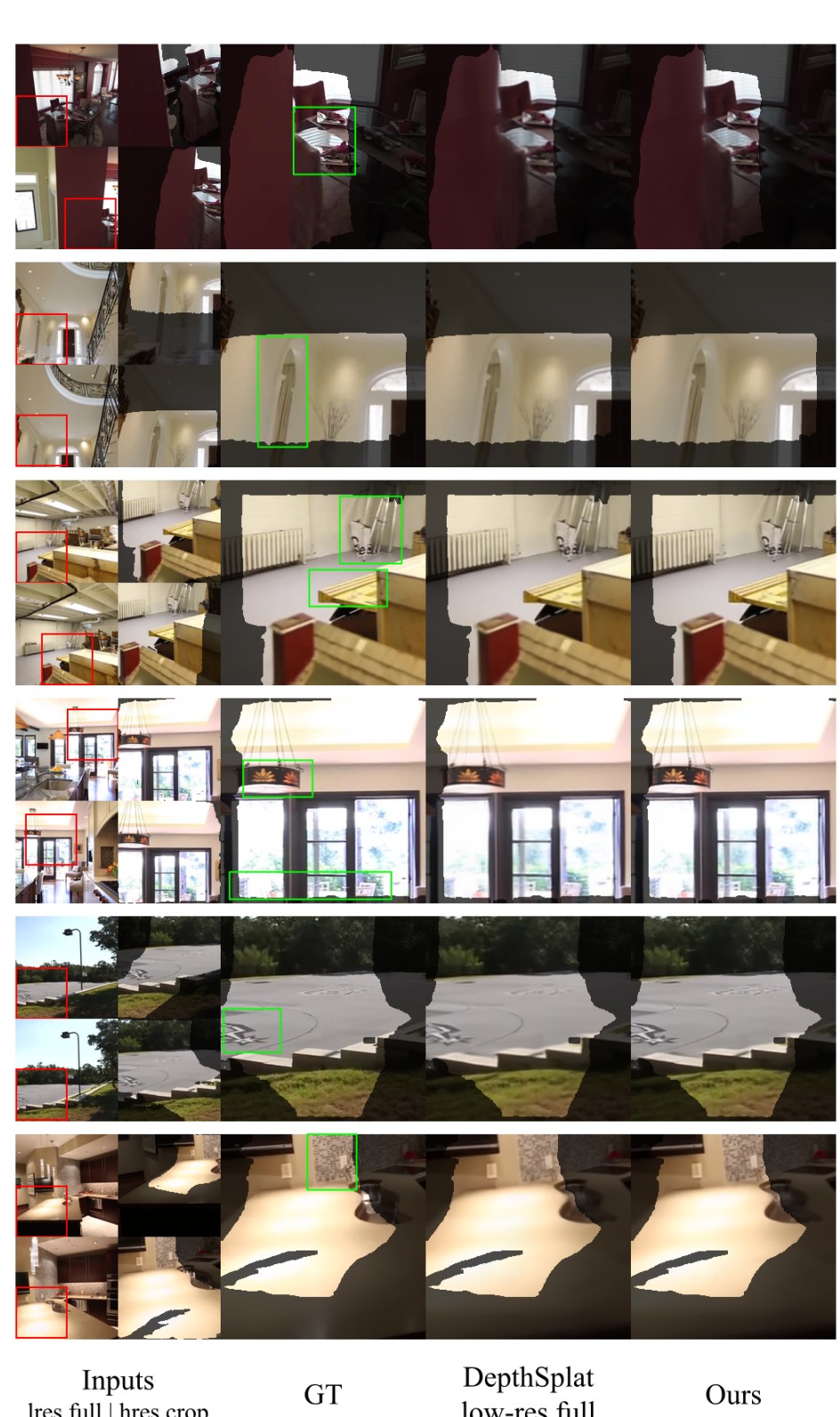

Inputs
lres full | hres crop

GT

DepthSplat
low-res full

Ours

Figure 8: **Novel view synthesis on RealEstate10K (Zhou et al., 2018).**

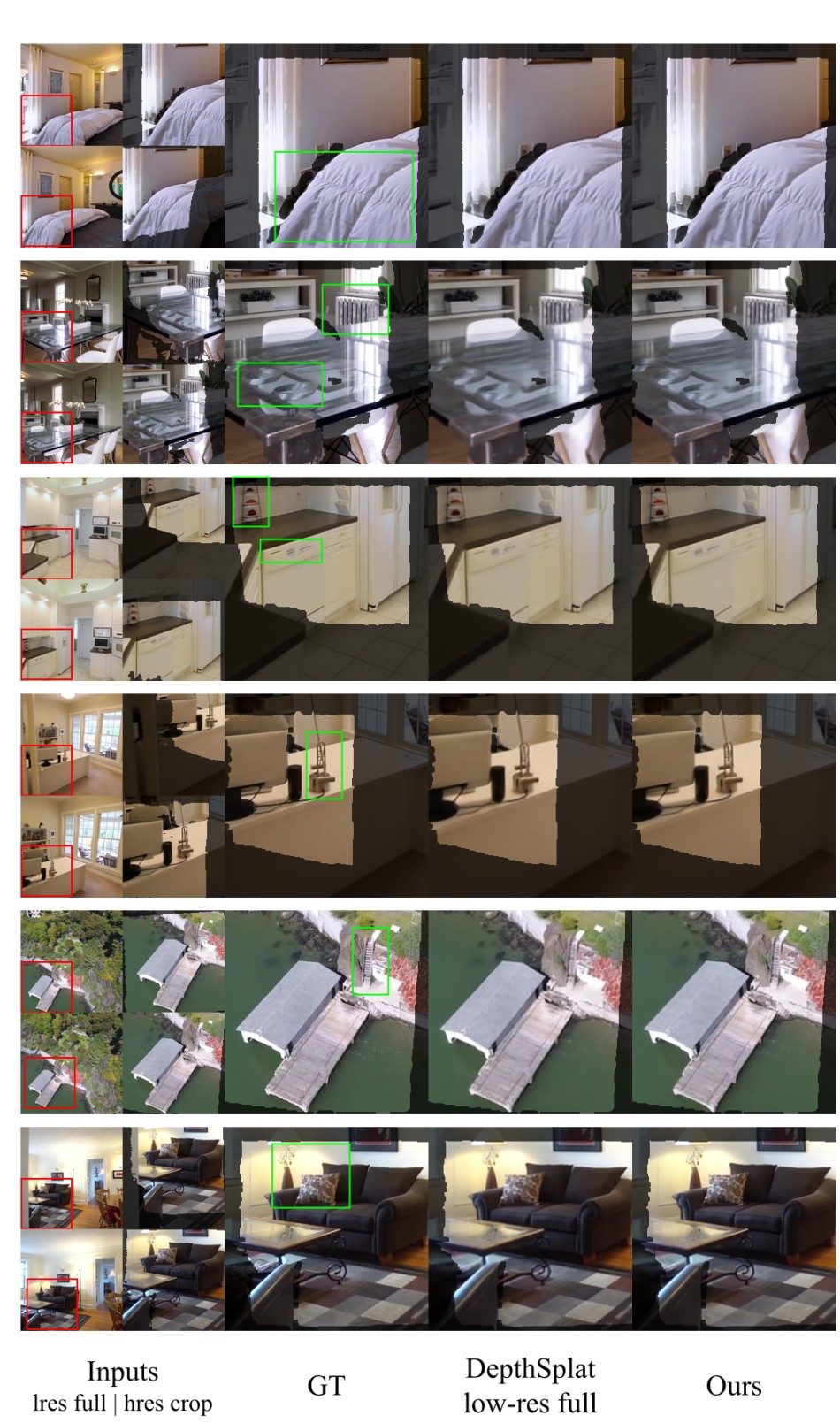

|  |  | DepthSplat |  |
| Inputs | GT | low-res full | Ours |
| lres full \| hres crop |  |  |  |

Figure 9: **Novel view synthesis on RealEstate10K (Zhou et al., 2018).**

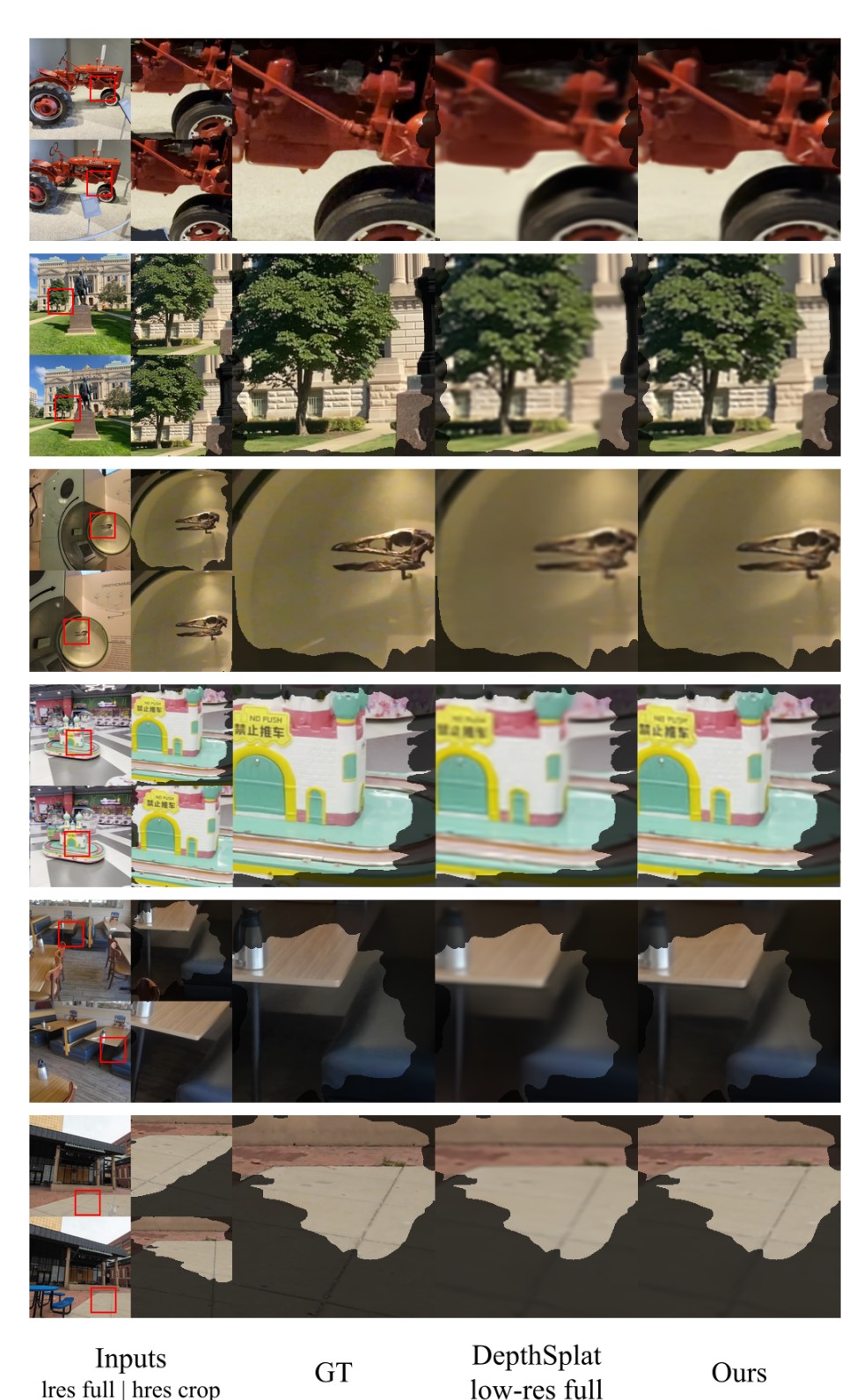

Inputs
lres full | hres crop

GT

DepthSplat
low-res full

Ours

Figure 10: **Novel view synthesis on DL3DV (Ling et al., 2024).**

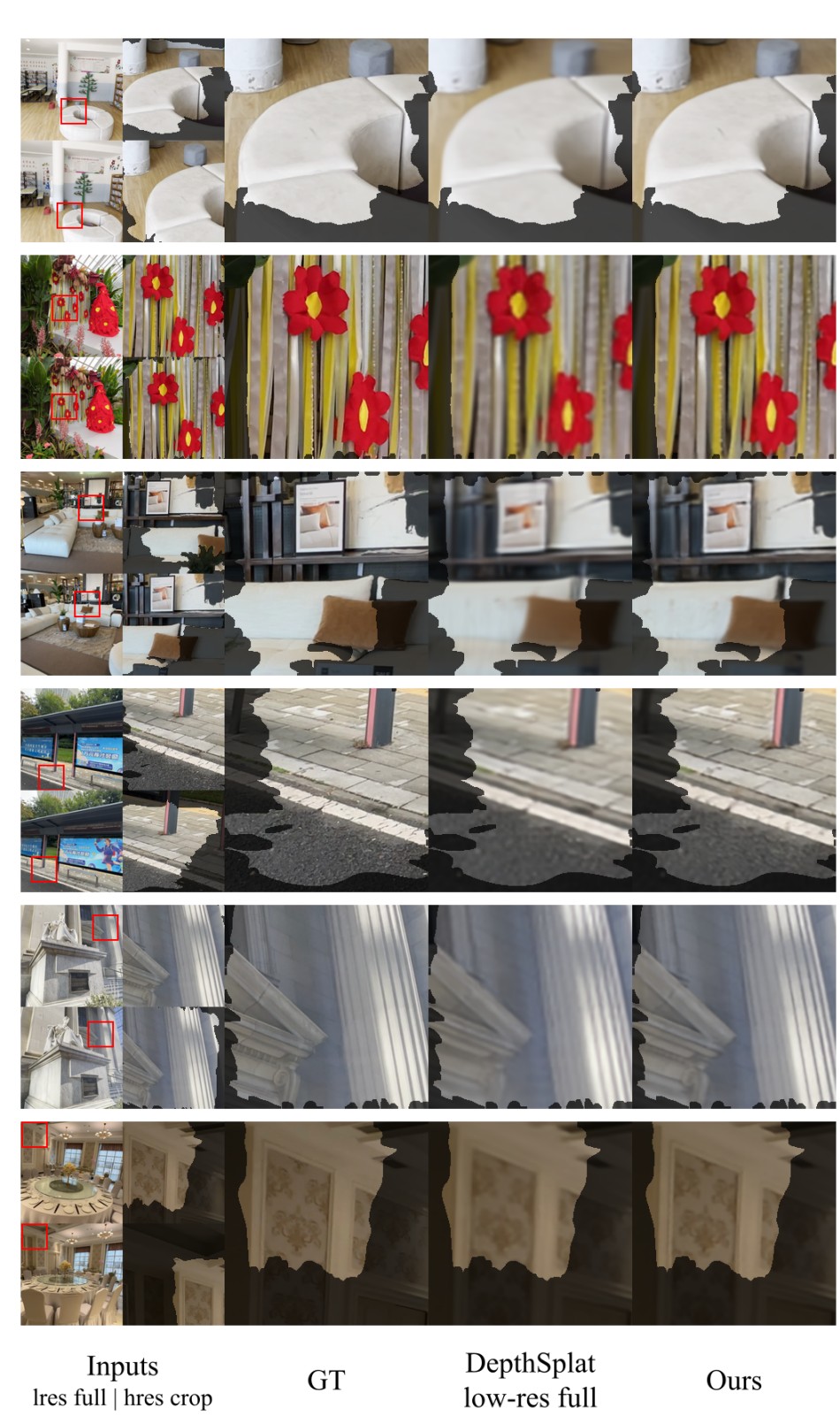

Inputs
lres full | hres crop

GT

DepthSplat
low-res full

Ours

Figure 11: **Novel view synthesis on DL3DV (Ling et al., 2024).**

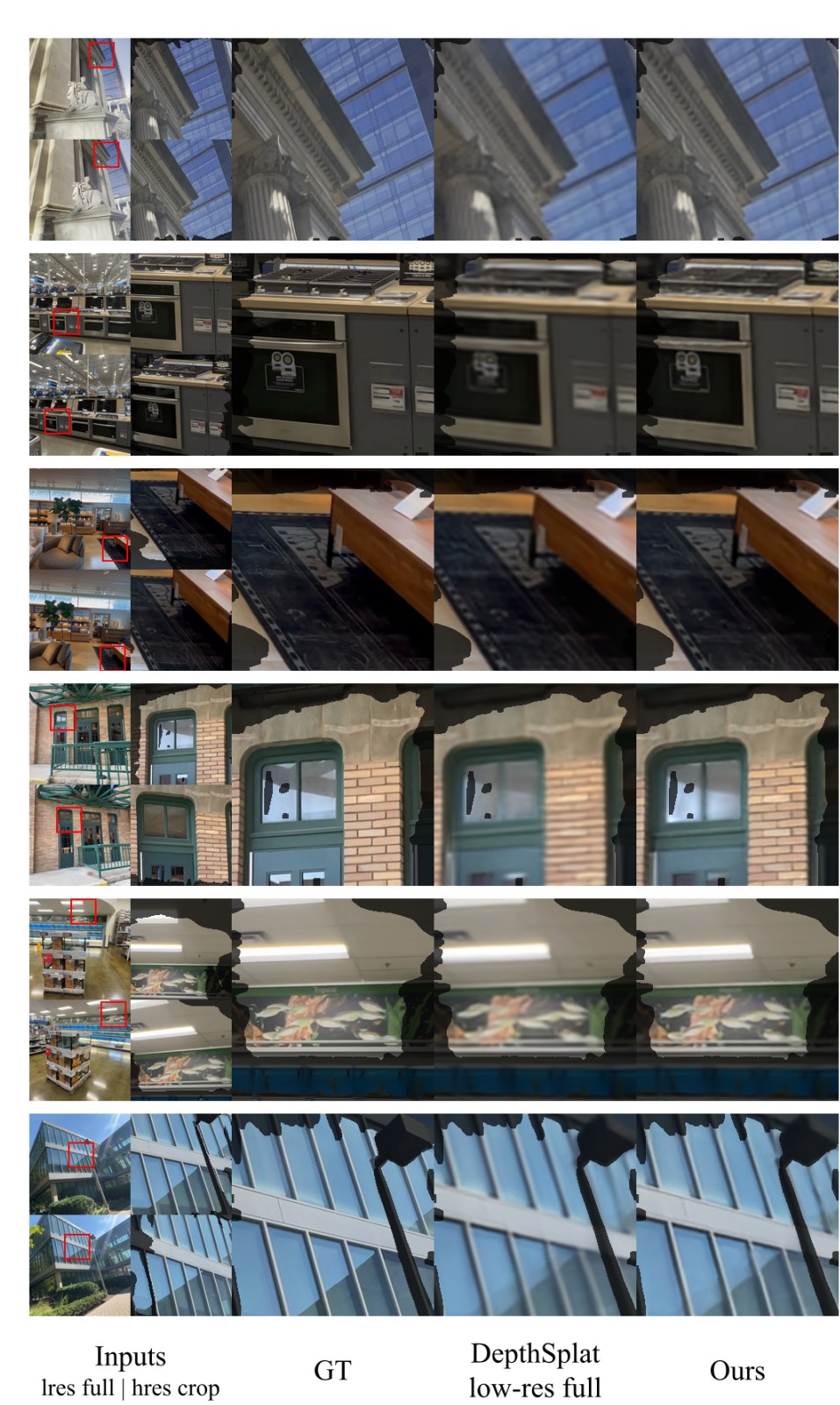

| Inputs | GT | DepthSplat | Ours |
| lres full \| hres crop | | low-res full | |

Figure 12: **Novel view synthesis on DL3DV (Ling et al., 2024).**

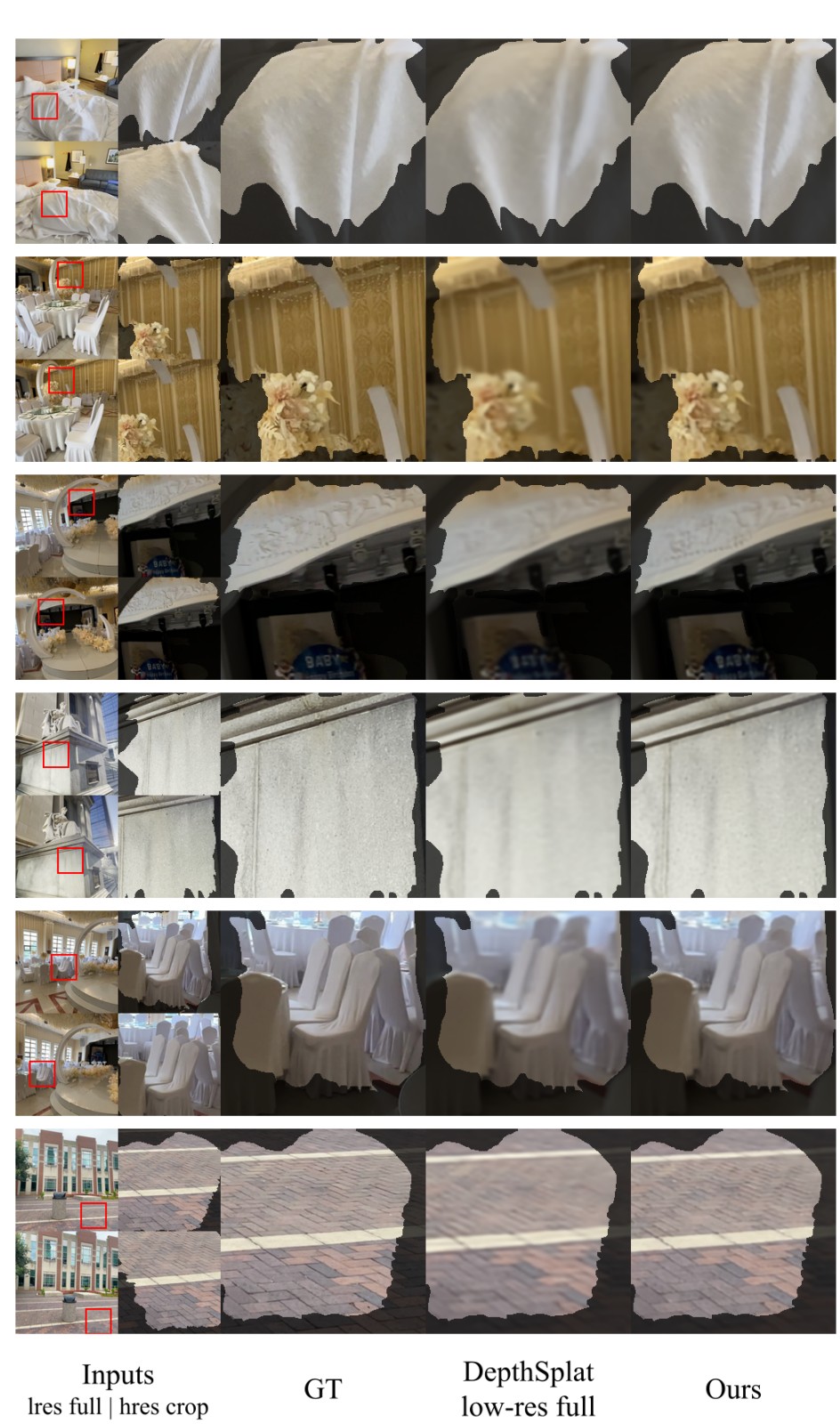

|  |  |  |  |
|---|---|---|---|
| Inputs
lres full \| hres crop | GT | DepthSplat
low-res full | Ours |

Figure 13: **Novel view synthesis on DL3DV (Ling et al., 2024).**

