# OpenReview forum: "GaussianLens: Localized High-Resolution Reconstruction via On-Demand Gaussian Densification"
_ICLR.cc/2026/Conference — Submitted to ICLR 2026_

### Official Review · Reviewer_neWX · 2025-10-30

**Soundness:** 3
**Presentation:** 3
**Contribution:** 2
**Rating:** 4
**Confidence:** 4

**Summary:**

This paper introduces GaussianLens, a framework for efficient, localized high-resolution 3D reconstruction. It avoids the cost of uniform high-resolution models by starting with a coarse scene and then densifying only a user-specified ROI.

**Strengths:**

1. The paper introduces a novel and practical problem: on-demand, localized high-resolution reconstruction. This aligns with real-world use cases and smartly addresses the efficiency gap between fast, low-detail generalizable models and slow, high-detail per-scene optimization methods.

2. The results are compelling, showing state-of-the-art quality with significantly lower computational cost.

3. The design is elegant. Pixel-guided densification provides an excellent high-frequency starting point, while the GaussianLens network with its efficient projection-based cross-attention intelligently refines these details into a coherent 3D structure. This directly and effectively tackles the core reconstruction challenge.

**Weaknesses:**

1. The paper fails to compare against a crucial baseline: using a feed-forward model for the coarse scene, then running per-scene optimization on the high-resolution ROI. This direct competitor might achieve higher quality, and its absence leaves the trade-off between GaussianLens's speed and optimization-based fidelity unevaluated.

2. The novelty of the projection-based cross-attention is overstated. The technique of projecting 3D points to sample 2D image features is a standard and widely used method in 3D vision. The paper presents this as a novel contribution while failing to cite extensive prior work, weakening its methodological claims.

**Questions:**

See Weaknesses

---

> ### Author Response · Authors · 2025-11-21
> **Author Response to Reviewer neWX**
>
> Thank you very much for your constructive review! We appreciate your positive remarks that our problem is novel and well-motivated, the results are compelling, and the design is elegant. We address your questions and concerns below.
>
> ## Q1. Per-Scene Optimization Baseline
> > 1.  The paper fails to compare against a crucial baseline: using a feed-forward model for the coarse scene, then running per-scene optimization on the high-resolution ROI. This direct competitor might achieve higher quality, and its absence leaves the trade-off between GaussianLens's speed and optimization-based fidelity unevaluated.
>
> Thanks for suggesting the important baseline. For both datasets, we take the coarse initial 3DGS predicted by DepthSplat, then optimize them with the high-resolution RoI context images. We use the state-of-the-art 3DGS implementation gsplat (https://github.com/nerfstudio-project/gsplat), and set the number of iterations so that the optimization time matches our model's inference time.
>
> - Our model's average inference time is $0.43$s on RE10K, and $0.40$s on DL3DV. We run gsplat per-scene optimization for **$30$** iterations on RE10K, which takes $0.41$s on average, for **$30$** iterations on DL3DV, which takes $0.39$s on average. Both methods are run on an a6000 GPU. To better understand the tradeoff between optimization time and quality, we also include per-scene optimization results at different number of iterations.
> (Note that we only count time in the training loop towards the optimization time and exclude the time for loading and evaluation. For reference, gsplat's reported average run time for $7000$ iterations on $4\times$ downsampled MipNeRF-360 (roughly $600\times800\sim800\times1200$ resolution) is 5min35s on A100, roughly $20$ iterations per second. Our iterations take even shorter time probably because the loss is only applied to a fewer number of in-RoI pixels.)
>
> - The table below shows results on RE10K. We achieve more than $2\times$ PSNR improvement with the same inference/optimization time. And it takes $3\times$ time for per-scene optimization to reach similar performance, demonstrating **our method's advantage in achieving high-quality results with fast speed**. We can also observe a slowdown in the PSNR improvement as the number of optimization steps increase, e.g. PSNR improves by $0.38$ from $30$ steps to $60$ steps, but $0.19$ from $60$ steps to $90$ steps, implying an upper bound on the performance of per-scene optimization. Specifically, in our sparse, two-context-view setting, per-scene optimization without learned priors may not be the most well-suited.
>
> | RE10K Results                      | Init (DepthSplat low-res, full) | Ours          | Optim. 30 Steps     | Optim. 40 Steps     | Optim. 50 Steps     | Optim. 60 Steps     | Optim. 70 Steps     | Optim. 80 Steps     | Optim. 90 Steps     |
> |-----------------------------|----------------------------------|----------------|----------------------|----------------------|----------------------|----------------------|----------------------|----------------------|----------------------|
> | PSNR                        | 27.28                            | **28.46(+1.18)**  | **27.78(+0.50)**        | 27.96(+0.68)        | 28.08(+0.80)        | 28.17(+0.88)        | 28.24(+0.96)        | 28.29(+1.01)        | **28.36(+1.07)**        |
> | Inference/Optimization Time (s) | /                                | **0.43**           | **0.41**                 | 0.54                 | 0.67                 | 0.81                 | 0.95                 | 1.07                 | **1.21**                 |
>
> (1/2)

---

> > ### Author Response · Authors · 2025-11-21
> >
> > (continued response to Q1)
> >
> > - The **deficiency of per-scene optimization under sparse view settings** is more pronounced on the DL3DV dataset, which contains more challenging scenes. As the table below shows, directly optimizing loss on the two context views leads to performance degradation on the target views. It takes $\sim100$ optimization steps and $4\times$ time to restore similar performance as the initial point. Beyond $100$ steps, the performance also oscillates rather than monotonically improving. Even with $300$ optimization steps and $9\times$ time, the PSNR improvement cannot reach half of our improvement. Note that training loss does go down in the optimization. We attribute the poor performance of per-scene optimization to the ill-posed sparse view setting, where prior-free optimization on sparse context views may lead to local minima. In contrast, our method effectively **leverages priors from training data and performs well under challenging sparse view settings.**
> >
> > | DL3DV Results               | Init (DepthSplat low-res, full) | Ours        | Optim. 30 Steps | Optim. 40 Steps | Optim. 60 Steps | Optim. 80 Steps | Optim. 100 Steps | Optim. 150 Steps | Optim. 200 Steps | Optim. 250 Steps | Optim. 300 Steps |
> > |----------------------|----------------------------------|-------------|------------------|------------------|------------------|------------------|-------------------|-------------------|-------------------|-------------------|-------------------|
> > | PSNR | 22.31                            | **23.62(+1.31)** | **20.51(-1.79)**     | 20.98(-1.31)     | 21.62(-0.68)     | 22.00(-0.30)     | 22.27(-0.03)      | 22.43(+0.13)      | 22.74(+0.44)      | 22.55(+0.25)      | 22.85(+0.55)      |
> > | Inference/Optimization Time (s)                | /                                | **0.40**         | **0.39**             | 0.52             | 0.76             | 1.01             | 1.26              | 1.85              | 2.45              | 3.04              | 3.63              |
> >
> >
> > ## Q2. Presentation of Projection-Based Cross-Attention
> >
> > > 2.  The novelty of the projection-based cross-attention is overstated. The technique of projecting 3D points to sample 2D image features is a standard and widely used method in 3D vision. The paper presents this as a novel contribution while failing to cite extensive prior work, weakening its methodological claims.
> >
> > - Thanks for the comment. We would like to clarify that we do not present this component as a novel contribution. In both the introduction and conclusion sections (quotes below), we only claim the **overall GaussianLens densification framework** as a contribution, which consists of feature initialization, the cross-modal transformer network, and the Gaussian decoder.
> > 	> (Introduction) ... We develop GaussianLens, a multi-modal framework featuring complementary mechanisms to fuse information from 3D Gaussians and multi-view images for effective densification prediction...
> >
> > 	> (Conclusion) ... We develop GaussianLens, a generalizable densification prediction framework that effectively fuses multi-modal input information...
> >
> > 	 While we don’t claim novelty for any individual component, we consider **the orchestration of these components**, designed and tailored for the image-conditioned local Gaussian densification problem, to be a meaningful contribution, besides the novel problem setting and the pixel-guided densification mechanism.
> >
> > -   We put the term "projection-based cross-attention" in bold and provided a separate illustration to **make it easier to reference and to clearly explain the mechanism**. Although “projection-based cross-attention” itself is not a new idea, we were unable to find a standard definition in prior work, as existing methods [1, 2] also adopt different implementations of the idea (e.g. some use direct projection and treat points as transparent ([1]), while others further consider occlusions and perform rasterization ([2])).
> >
> > -   We will **add citations** to works that use similar projection-based point cloud-image attentions (e.g. [1, 2]).
> >
> > -   We will **tone down any wording** that may have unintentionally suggested a novelty claim. We previously used “employ”, ”introduce”, and “propose” when describing “projection-based cross-attention”. We will revise the latter two to more neutral verbs such as “use” or “employ”.
> >
> > [1] Li et al. "Deepfusion: Lidar-camera deep fusion for multi-modal 3d object detection." Proceedings of the IEEE/CVF conference on computer vision and pattern recognition. 2022.
> > [2] Lu et al. "Large point-to-gaussian model for image-to-3d generation." Proceedings of the 32nd ACM International Conference on Multimedia. 2024.
> >
> > (2/2)

---

### Official Review · Reviewer_4dun · 2025-11-01

**Soundness:** 2
**Presentation:** 3
**Contribution:** 3
**Rating:** 6
**Confidence:** 4

**Summary:**

This paper proposes a method to densify and refine initial low-resolution Gaussians to reconstruct local regions of interest (RoIs) in high resolution on demand. The refinement process is guided by cropped regions of high-resolution images, eliminating the need to process entire images and thereby reducing computational cost. Additionally, a pixel-guided densification strategy is introduced to prevent the initial Gaussians from over-covering pixels within high-resolution RoIs, leading to improved reconstruction quality.

**Strengths:**

- The paper is well written, and both the methodology and experimental setups are clearly and thoroughly explained.
- The ablation study is comprehensive, and each component of the proposed framework is empirically validated.
- The proposed method is flexible and can be applied to various sources of initial Gaussians, whether generated by feed-forward or optimization-based models.

**Weaknesses:**

- The proposed approach reconstructs the scene once using low-resolution full images and subsequently refines specific RoIs using cropped high-resolution images. While this strategy aims to reduce the memory and computational costs of full high-resolution reconstruction, its practicality is mainly justified in scenarios with extremely high-resolution inputs or a large number of views that exceed the memory budget. However, recent works such as Long-LRM and LVT [1] have introduced techniques that significantly reduce the computation of multi-view attention—the main bottleneck in feed-forward models. In particular, LVT [1] demonstrates the ability to reconstruct 2K-resolution scenes, which raises questions about the necessity and advantage of the proposed approach in such cases.
- The proposed framework requires an additional inference step from the fine network each time a new RoI is reconstructed. In contrast, conventional feed-forward Gaussian models generate the entire set of Gaussians in a single forward pass, which can be potentially expensive initially but does not require additional inference thereafter. In the proposed method, while the coarse network inference is relatively lightweight due to lower-resolution inputs, the repeated fine-network inferences for each RoI could introduce notable overhead. This repeated computation may limit the practicality of the method in real-world applications where frequent RoI updates are required.

[1] LVT: Large-Scale Scene Reconstruction via Local View Transformers, SIGGRAPH Asia, 2025.

**Questions:**

- Can the proposed method accept 3D RoIs instead of 2D ones? It seems somewhat unrealistic to assume that RoIs are provided as 2D binary masks. In practical scenarios, selecting RoIs directly from the initial 3D reconstruction and projecting them onto each input view would be more reasonable. Could the authors clarify how the method could handle such a setup?

---

> ### Author Response · Authors · 2025-11-21
> **Author Response to Reviewer 4dun**
>
> Thank you very much for your insightful and detailed review! We appreciate your positive remarks on the clarity of the paper, the comprehensiveness of the ablation study, and the flexibility of the proposed method. We address your questions below.
>
> ## Q1. Comparison to Efficient Feed-Forward Models
> > 1.  The proposed approach reconstructs the scene once using low-resolution full images and subsequently refines specific RoIs using cropped high-resolution images. While this strategy aims to reduce the memory and computational costs of full high-resolution reconstruction, its practicality is mainly justified in scenarios with extremely high-resolution inputs or a large number of views that exceed the memory budget. However, recent works such as Long-LRM and LVT [1] have introduced techniques that significantly reduce the computation of multi-view attention—the main bottleneck in feed-forward models. In particular, LVT [1] demonstrates the ability to reconstruct 2K-resolution scenes, which raises questions about the necessity and advantage of the proposed approach in such cases.
>
> Thanks for bringing up the comparison to recent, more efficient feed-forward models. We acknowledge the exciting progress producing high-resolution outputs with more efficient feed-forward models. However, we believe our method offers distinct advantages in both efficiency and application flexibility. It can also benefit from the advances in feed-forward models.
>
> -   Regarding **computational efficiency**.
> 	- Our method reduces cost and redundancy by **reconstructing fewer targets**, while feed-forward methods **reconstruct the same targets with less compute**. Specifically, our method concentrates the computational budget exclusively on the regions of interest, and saves the redundancy in reconstructing regions of less importance. We consider our method and efficient feed-forward models to be **parallel and complementary approaches to improving efficiency**.
> 	- Our computational cost **scales with the number of pixels in the high-resolution RoI**. There is **no theoretical limit on the original resolution** of the full-size capture.
> 		- As shown in Supp.A.2, our method can handle inputs at $2048\times 2048$, which exceeds Long-LRM’s $960\times 540$ (on par if they shift their sequence length $32$ to spatial resolutions), and LVT’s “2K” (“2K” in DL3DV’s official dataset download link corresponds to $1920\times1080$; LVT was also published after the ICLR submission deadline).
> 		- Long-LRM reduces cost by combining linear-cost Mamba2 with transformer blocks, enabling a larger number of tokens (which scales linearly with the sequence length but quadratically with the image resolution). LVT reduces cost by restricting cross-view attention to local neighborhoods, which improves scaling to the number of views.  Both still **require engineering to reduce cost at the spatial dimension**. Long-LRM prunes away a fixed percentage of lowest opacities and merges spatially adjacent $2\times2$ patch tokens; LVT achieves 2K resolution with specifically engineered larger patch tokens, suggesting **an engineering upper bound for global, full frame reconstructions**.
> 	- In addition to reduced computation during training and inference, our method also reconstructs with **fewer Gaussians**, **which saves the cost of storage, data transfer, and rendering**. LVT also mentions producing "dence, redundant representation" and the resulting "splat rasterization can be come a significant bottleneck" as their limitation.
> 	-  Our approach is **can benefit from** the progress in more efficient multi-view attention. For a concrete example, we encode multi-view image features with a pretrained multi-view transformer-based feature extractor from DepthSplat. And the encoding process takes up a considerable portion of our inference time. More efficient multi-view attention can potentially reduce that time.
>
>
>
> - Our model follows the **refinement** paradigm, while feed-forward models perform **single pass prediction**.
> 	- From a high level, the ability to **improve an existing prediction** is fundamentally different from and complementary to the ability to **predict accurately in a single pass**. Our method **incorporates new observations into an existing reconstruction by adding local details**, enabling **incremental updates that build upon prior results**.
> 	- This refinement paradigm is **uniquely useful** in cases when
> 		- New observations arrive over time, e.g. in interactive reconstruction, when the user decides to approach an RoI and take new close-up captures.
> 		- The scene complexity makes single-pass prediction challenging, e.g. super-high resolution inputs as mentioned in the discussion.
>
> (1/2)

---

> > ### Author Response · Authors · 2025-11-21
> >
> > ## Q2. Efficiency when Reconstructing Multiple RoIs
> > > 2.  The proposed framework requires an additional inference step from the fine network each time a new RoI is reconstructed. In contrast, conventional feed-forward Gaussian models generate the entire set of Gaussians in a single forward pass, which can be potentially expensive initially but does not require additional inference thereafter. In the proposed method, while the coarse network inference is relatively lightweight due to lower-resolution inputs, the repeated fine-network inferences for each RoI could introduce notable overhead. This repeated computation may limit the practicality of the method in real-world applications where frequent RoI updates are required.
> >
> >
> > Thank you for raising the potential efficiency problem in repeated inference on new RoIs.
> > - Our **average network inference time is $0.43$s on RE10K and $0.40$s on DL3DV**. If we consider interactive scenarios where RoIs are sequentially specified by human users, such inference speed should be able to **match human's reaction time** (according to [1](https://taylorandfrancis.com/knowledge/Medicine_and_healthcare/Psychiatry/Reaction_time/), average human's reaction time to a visual stimulus is typically $0.25$s.) plus the time required to specify new RoIs. Therefore, we believe the method remains **practical in real-world applications**. As we perform local updates, there is also potential to parallelize the reconstruction of non-overlapping RoIs to further reduce cost, e.g. when multiple users interact with the same virtual scene.
> > - **Conventional feed-forward networks struggle to scale to very high-resolution inputs**, e.g., DL3DV fails to train on $1024\times1024$ inputs on an H100 GPU. In such cases, our method **enables the use of high-resolution observations** that would otherwise be impossible to leverage. In other words, our improvement in efficiency is **not only an incremental reduction of time and memory, but makes a step-change that enables high reconstruction resolutions not feasible with feed-forward models**.
> > - When the input resolution is moderately low and can be handled by a feed-forward model, the choice between the one-time global inference of the feed-forward model and the per-RoI inference of our method should depend on the exact use case. When the number of RoI inferences go beyond a certain threshold, running the feed-forward model once may lead to shorter total time.
> >
> >
> > ## Q3. Handling RoIs Specified in 3D
> > > 3.  Can the proposed method accept 3D RoIs instead of 2D ones? It seems somewhat unrealistic to assume that RoIs are provided as 2D binary masks. In practical scenarios, selecting RoIs directly from the initial 3D reconstruction and projecting them onto each input view would be more reasonable. Could the authors clarify how the method could handle such a setup?
> >
> >
> > Yes. Directly specifying 3D RoIs is in fact more straightforward for our method.
> > Given user-specified RoIs, our method computes a binary mask over input Gaussians to determine which Gaussians are in the region, so that we can densify from them.
> > - When RoIs are given in 2D, we unproject them to 3D as described in Supp. B. 1. Region-of-Interest Gaussian Selection.
> > - When RoIs are given in 3D, we can treat each Gaussian as an ellipsoid (by thresholding its density), and directly compute whether it intersects with the 3D region, whether the region is given as voxels, unions of 3D balls, etc. If the selection happens directly on the Gaussians, it gives the binary mask already.
> >
> > (2/2)

---

### Official Review · Reviewer_3Ug4 · 2025-11-01

**Soundness:** 3
**Presentation:** 3
**Contribution:** 3
**Rating:** 4
**Confidence:** 4

**Summary:**

This paper introduces GaussianLens, a feed-forward framework for localized high-resolution 3D reconstruction that densifies a coarse, low-resolution 3D Gaussian Splatting (3DGS) result within a user-specified Region of Interest (RoI). Starting from an initial low-res reconstruction (e.g., from DepthSplat), the method fuses multi-modal cues, rendering residuals, gradients, and multi-view image features via a PointTransformerV3-based encoder with projection-based image and Gaussian cross-attention, then predicts residual updates to generate a densified set of Gaussians in the RoI. To handle large zoom factors, it further proposes pixel-guided densification, spawning per-pixel Gaussians in the RoI using back-projected depths from the coarse model to preserve fine details. On benchmarks built from RealEstate10K and DL3DV, GaussianLens improves RoI view-synthesis quality over low-res baselines, matches or exceeds uniform high-res models with substantially fewer Gaussians.

**Strengths:**

1. Good writing, with clear organization and methodological description.

2. Engineering is solid and the effectiveness is strong.

3. The motivation is clear, and the proposed modules are appropriate for the current pipeline.

**Weaknesses:**

1. The pipeline for solving the problem is overly redundant and too engineering-heavy. The paper targets 3D reconstruction of RoI regions under limited resources and high-resolution inputs. However, it employs a two-stage scheme, which first uses DepthSplat to produce a low-resolution 3DGS, then refines and supplements it with high-resolution images. While this leverages the strong capabilities of existing networks like DepthSplat and PointTransformerv3, it may be redundant and inefficient. In principle, a more novel and natural approach would be a single-stage design that performs interaction between the RoI in high-resolution images and the correlated regions in the global low-resolution image during DepthSplat inference, directly inferring Gaussians with different densities rather than post-processing. I consider this the core issue: the foundational pipeline contains substantial redundancy, making the work more incremental and significantly weakening its novelty.

2. The final loss only applies to the RoI, which continually optimizes Gaussians inside the RoI while ignoring background Gaussians. Because RoI Gaussians also influence background areas, especially transition regions, this may enhance the RoI but degrade the background.

3. In real applications, the user’s RoI on a single image is first mapped onto the 3D scene generated by DepthSplat. However, this scene itself may be inaccurate, especially with depth outliers. This can make the corresponding RoI at another viewpoint deviate significantly, leading to performance degradation.

**Questions:**

See the weaknesses.

---

> ### Author Response · Authors · 2025-11-21
> **Author Response to Reviewer 3Ug4**
>
> Thank you very much for your thoughtful and constructive review! We appreciate your positive remarks on the clarity of our motivation and the effectiveness of our approach. And we address your questions and concerns below.
>
>
> ## Q1. Problem Setting, Pipeline Redundancy, and Novelty
> > 1.  The pipeline for solving the problem is overly redundant and too engineering-heavy. The paper targets 3D reconstruction of RoI regions under limited resources and high-resolution inputs. However, it employs a two-stage scheme, which first uses DepthSplat to produce a low-resolution 3DGS, then refines and supplements it with high-resolution images. While this leverages the strong capabilities of existing networks like DepthSplat and PointTransformerv3, it may be redundant and inefficient. In principle, a more novel and natural approach would be a single-stage design that performs interaction between the RoI in high-resolution images and the correlated regions in the global low-resolution image during DepthSplat inference, directly inferring Gaussians with different densities rather than post-processing. I consider this the core issue: the foundational pipeline contains substantial redundancy, making the work more incremental and significantly weakening its novelty.
>
> Thanks for the thoughtful and inspiring comment.
> - We would like to first clarify our problem setting.
> We focus on the Gaussian local densification problem, where **the low-resolution 3DGS is given as part of the input. Producing the low-resolution 3DGS is not a part of our method.** Our method instead performs a **single-stage refinement, mapping low-res Gaussians and high-res images to high-res Gaussians**.
>
> - This setting is **flexible** with respect to the source of the initial Gaussians.
> 	- Low-resolution Gaussians can be easily obtained with feed-forward models such as DepthSplat/MVSplat/pixelSplat, or from lower-cost versions of per-scene optimization using low-res images and fewer optimization steps. The choice depends on the application scenario.
> 	- **Our method can be applied to low-resolution Gaussians from various sources**, as shown in Sec. 4.4, without the need to adapt the network design specifically to each source, jointly train with the source model, or access the internal features of the source model. It can be seen as a plug-and-use module to enhance the detail reconstruction of any existing method. With the rapid development of state-of-the-art feed-forward models, we expect our approach to **remain broadly compatible and to benefit directly from future advances in feed-forward Gaussian prediction.**
> - This setting also **fits the use case where users determine the region of interest based on a coarse overall understanding of the scene**. The coarse initial reconstruction is first presented to the user as a global context, based on which specific regions of interest can be determined, and further refined by our method.
>
> - The **novelty** of our method also lies in this setting.
> 	- From a high level, the ability to **improve an existing prediction** is fundamentally different from and complementary to the ability to **predict accurately in a single pass**. Specifically, our method **incorporates new observations into an existing reconstruction by adding local details**, enabling **incremental updates that build upon prior results**.
> 	- This refinement paradigm is **uniquely useful** in many cases. For example, when 1) new observations arrive over time, 2) the target focus (RoI) of reconstruction changes over time, and 3) the scene complexity makes single-pass prediction challenging. However, **the refinement-based paradigm remains under-explored** compared to single-pass direct prediction, with only a few pioneering works exploring similar ideas (e.g. Chen et al. (2024b), Chen et al. (2024c)).
> 	- This paradigm also **poses unique technical challenges** that our method addresses with novel designs. For example, it requires **understanding and modeling of residual signals** between the current prediction and the target. It also requires **cross-modality information fusion** from both 3D Gaussians and 2D images. We tackle these challenges with the GaussianLens cross-modal Gaussian densification framework and pixel-guided Gaussian densification mechanism, both effectively associating input Gaussians and images and capturing the residuals between them.
>
> (1/4)

---

> > ### Author Response · Authors · 2025-11-21
> >
> > (continued response to Q1)
> > -  The proposed design where high-resolution RoI and low-resolution global images interact in a DepthSplat-like multi-view transformer framework is very interesting. In fact, we did consider this approach before. We believe it has the potential to produce better geometry by applying the strong geometric reasoning ability of DepthSplat to the additional high-resolution images supplied. Whether or not to explicitly build and use a low-res 3D reconstruction is also an interesting research question. It may serve as a useful scaffold to add detail to, as the comparison between our method and `DepthSplat, high-res crop` suggests. It may also introduce undesirable noises when itself is inaccurate. However, we consider this **a parallel direction focused on single-pass direct prediction**, rather than a refinement method capable of improving existing reconstructions from flexible sources. Compared to us, it has different settings (image to Gaussian v.s. Gaussian + image to Gaussian) and application cases (pre-specified image RoIs v.s. dynamically specified RoIs given low-res overall reconstructions and dynamically arriving observations).
> >
> > (2/4)

---

> > > ### Author Response · Authors · 2025-11-21
> > >
> > > ## Q2. Preserving Background Reconstruction Quality
> > > > 2.  The final loss only applies to the RoI, which continually optimizes Gaussians inside the RoI while ignoring background Gaussians. Because RoI Gaussians also influence background areas, especially transition regions, this may enhance the RoI but degrade the background.
> > >
> > > Thanks for bringing the potential problem to discussion. We agree that preserving the background content while enhancing the region of interest is very important in applications.
> > >
> > > - **Currently, the background quality is largely preserved.** In our reconstruction, the PSNR of the background region **degrades by an almost negligible 0.033 dB**. In the table below, we take the images rendered from our output Gaussians on DL3DV, and compute the masked metrics over 1) all image pixels, 2) pixels in the RoI region (reported in the main paper), and 3) pixels outside the RoI region (background). For a comprehensive evaluation, we include results for both the $256\times256$ crop containing the RoI, and the $1024\times1024$ full image (where both improvement and degradation can be diluted as the RoI only takes up a small portion of the image). Our results achieve significant improvement in the region of interest, with almost negligible degradation in the background.
> > >
> > > | Evaluation Region                              | Initial PSNR | Our PSNR | PSNR Difference | Initial SSIM | Our SSIM | SSIM Difference |
> > > |------------------------------------------------|--------------|----------|------------------|--------------|----------|------------------|
> > > | Pixels in RoI mask                             | 22.31        | 23.62    | 1.312            | 0.652        | 0.719    | 0.067            |
> > > | 256x256 RoI Image Crop - All Pixels            | 22.02        | 22.86    | 0.845            | 0.648        | 0.701    | 0.053            |
> > > | 256x256 RoI Image Crop, Pixels out of RoI mask       | 21.84        | 21.81    | -0.033           | 0.650        | 0.648    | -0.002           |
> > > | 1024x1024 Full Image - All Pixels              | 21.64        | 21.68    | 0.037            | 0.687        | 0.690    | 0.004            |
> > > | 1024x1024 Full Image, Pixels out of RoI mask         | 21.65        | 21.64    | -0.013           | 0.689        | 0.689    | 0.000            |
> > >
> > > - **Our current 2D RoI masks cover most of the region that may be influenced by RoI Gaussians.** We compute our 2D RoI masks by thresholding the rendered alpha channel from RoI Gaussians, and the threshold is set to a small value of $\tau = 0.1$. They cover most of the regions that may be significantly influenced by the RoI Gaussians.
> > > - **To further improve background quality preservation, we can enlarge the region we apply loss to,** by 1) decreasing the alpha threshold $\tau$, and 2) computing a "transition region mask" by dilating the original 2D RoI masks by a few pixels, and imposing a "transition region" loss on the transition region.
> > >
> > > (3/4)

---

> > > > ### Author Response · Authors · 2025-11-21
> > > >
> > > > ## Q3. Handling RoIs Specified from a Single Image
> > > > > 3.  In real applications, the user’s RoI on a single image is first mapped onto the 3D scene generated by DepthSplat. However, this scene itself may be inaccurate, especially with depth outliers. This can make the corresponding RoI at another viewpoint deviate significantly, leading to performance degradation.
> > > >
> > > > Thank you for raising this point. We acknowledge that this is a potential problem. As a refinement/densification method, error in the initial scene reconstruction is a critical factor for us. We analyze the problem in two folds. 1) How well we can handle and correct errors in the initial reconstruction in general. 2) How we may handle specific noise coming from `single-view RoI unprojection to inaccurate initial Gaussians`.
> > > >
> > > > - Handle and correct errors in the initial reconstruction in general.
> > > > 	- Our results on DL3DV and RE10K with initial scenes reconstructed by DepthSplat, as well as worse-performing MVSplat and pixelSplat **demonstrate our method's robustness to common errors present in initial reconstructions**.
> > > > 	- **Handling and correcting catastrophic errors from the initial scene reconstruction is out of the scope of our method**. As we mentioned in the conclusion, our model focuses on improving details based on the initial coarse reconstruction. It is not designed to handle catastrophic errors already present in the initial reconstruction.
> > > >
> > > >
> > > > - Handle noisy RoI Gaussian selection as a result of single-view RoI unprojection.
> > > > 	- Compared to our standard evaluation setting, where we only take Gaussians $\mathcal{G}^{2-view}$ visible from both context view RoIs, single-view-based selection $\mathcal{G}^{1-view}$ will include more Gaussians, which we denote by $\mathcal{G}^{noise} = \mathcal{G}^{1-view} \backslash \mathcal{G}^{2-view}$. **We can train the network to ignore the additional Gaussians $\mathcal{G}^{noise}$, by finetuning it on $\mathcal{G}^{1-view}$ instead of $\mathcal{G}^{2-view}$ as the input Gaussians.**
> > > > 	- Single-view RoI will also reduce our pixel-guided Gaussians $\mathcal{G}^{pixel} = \mathcal{G}^{v1}$ $\cup$ $\mathcal{G}^{v2}$ to only come from one view, i.e. $\mathcal{G}^{v1}$. We can either
> > > > 		- finetune the network on pixel-guided Gaussians from one view only, i.e. on $\mathcal{G}^{v1}$
> > > > 		- or compute pseudo RoI masks by projecting the RoI Gaussians $\mathcal{G}^{1-view}$ to the other view, use pseudo masks to generate pixel-guided Gaussians $\hat{\mathcal{G}^{v2}}$, then finetune the network on $\mathcal{G}^{v1} \cup \hat{\mathcal{G}^{v2}}$.
> > > > 	- Note that although the RoI is specified from a single view, the initial reconstruction (e.g. from DepthSplat) still comes from at least two views, and the quality of the reconstruction should remain the same. Given the method's ability to handle general noise in the current initial reconstruction, we expect it can also learn to ignore outlier Gaussians introduced by single-view unprojection.
> > > >
> > > > (4/4)

---

> > > > > ### Comment · Reviewer_3Ug4 · 2025-11-28
> > > > >
> > > > > Thanks for your response. After reading the above clarification and support statement, I admit the novelty of GaussianLens. Although it may not be the optimal setting when given both low- and high-resolution pictures, it indeed can be used in existing feed-forward GS methods easily, which demonstrates its superior generalization. Meanwhile, although there are some issues, such as background degradation and inaccurate 3D RoIs, the overall performance is acceptable. I hope to see the plus version of GaussianLens to solve these problems. Considering that, I support the acceptance of this submission version.

---

### Meta-Review · Area_Chair_2kg7 · 2026-01-07

**Summary:**

The initial ratings for this paper are 2x marginal rejects and 1x marginal accept. The strengths of the paper are: 1) clear motivation; 2) good writing; 3) proposed method is flexible and can be applied to  can be applied to various sources of initial Gassians; 4) ablations are comprehensive. However, there are major weaknesses pointed out by the reviewers: 1) pipeline for solving the problem is overly redundant and too engineering-heavy; 2) the method is not robust to outliers and thus might be limited in real applications. 3) paper fails to compare against a crucial baseline: using a feed-forward model for the coarse scene, then running per-scene optimization on the high-resolution ROI. 4) novelty of the projection-based cross-attention is overstated. Although one of the reviewers who initially gave a marginal reject is agreeable to raise the score to accept, there are still remaining concerns in the final comment on background degradation and inaccurate 3D RoIs. The doubts on overstating the contribution and failure to compare with important baseline also still remain. Since the weaknesses outweigh the strengths of the paper, the AC decides to reject the paper.

**Reviewer Concerns:**

there are major weaknesses pointed out by the reviewers: 1) pipeline for solving the problem is overly redundant and too engineering-heavy; 2) the method is not robust to outliers and thus might be limited in real applications. 3) paper fails to compare against a crucial baseline: using a feed-forward model for the coarse scene, then running per-scene optimization on the high-resolution ROI. 4) novelty of the projection-based cross-attention is overstated. Although one of the reviewers who initially gave a marginal reject is agreeable to raise the score to accept, there are still remaining concerns in the final comment on background degradation and inaccurate 3D RoIs. The doubts on overstating the contribution and failure to compare with important baseline also still remain.

**Reviewer Scores:**

One reviewer will maintain marginal reject. The other one who initially gave a marginal reject mentioned a raise of the score, but commented there are still some issues with the work. Thus, this would probably be a borderline accept and would still be okay to reject. The other reviewer would maintain the marginal accept.

---

### Decision · Program_Chairs · 2026-01-26

Reject